# Combinatorial encoding of odors in the mosquito antennal lobe

Pranjul Singh [1], Shefali Goyal[1], Smith Gupta [1], Sanket Garg [1,2], Abhinav Tiwari [1], Varad Rajput [1], Alexander Shakeel Bates [3], Arjit Kant Gupta [1] & Nitin Gupta [1,4] ✉

Among the cues that a mosquito uses to find a host for blood-feeding, the smell of the host plays an important role. Previous studies have shown that host odors contain hundreds of chemical odorants, which are detected by different receptors on the peripheral sensory organs of mosquitoes. But how individual odorants are encoded by downstream neurons in the mosquito brain is not known. We developed an in vivo preparation for patch-clamp electrophysiology to record from projection neurons and local neurons in the antennal lobe of *Aedes aegypti*. Combining intracellular recordings with dye-fills, morphological reconstructions, and immunohistochemistry, we identify different sub-classes of antennal lobe neurons and their putative interactions. Our recordings show that an odorant can activate multiple neurons innervating different glomeruli, and that the stimulus identity and its behavioral preference are represented in the population activity of the projection neurons. Our results provide a detailed description of the second-order olfactory neurons in the central nervous system of mosquitoes and lay a foundation for understanding the neural basis of their olfactory behaviors.

*Aedes aegypti* has evolved as an anthropophilic mosquito species and a global vector of multiple diseases including dengue, Zika fever, and chikungunya[1,2]. Along with temperature, humidity, and visual cues, olfaction plays a crucial role in the host-seeking behavior of mosquitoes[3–7]. Volatile chemical molecules released from skin or present in breath guide mosquitoes toward their host[8–11]. Like other insects, mosquitoes also use olfaction for finding food, mates, and oviposition sites[12–18].

The main olfactory organs in mosquitoes, like most insects, include the antennae and the maxillary palps[9,19]. The axons of the sensory neurons project to the antennal lobe (AL)[20], where they synapse with broadly two classes of second-order olfactory neurons: projection neurons (PNs) and local neurons (LNs)[21]. The insect AL, analogous to the olfactory bulb in vertebrates, is compartmentalized into glomeruli, such that each glomerulus receives input from a specific type of receptor neurons[22–26]. Although very little is known about the numbers and the morphologies of LNs and PNs in mosquitoes[21], studies on other insects have shown that insect AL typically contains a few hundred LNs and PNs[27–31]. LNs innervate multiple glomeruli and provide lateral interactions within the AL[27,32–34]. PNs innervate one or a few glomeruli and carry the output of the AL to the mushroom body and the lateral horn[28].

How are odors encoded by the PNs? Prior work in other insects suggests a combinatorial code, with each odor activating multiple PNs[35,36], which emerges from combinatorial activity at the level of the receptors[37–39] along with lateral interactions within the AL. The vertebrate olfactory system also uses a combinatorial code for odors[40]. While a combinatorial code has obvious advantages for representing a large number of odors and for flexibly associating them with approach/avoidance behaviors[41–43], how it preserves innate preferences to

[1]Department of Biological Sciences and Bioengineering, Indian Institute of Technology Kanpur, Kanpur, Uttar Pradesh 208016, India. [2]Department of Economic Sciences, Indian Institute of Technology Kanpur, Kanpur, Uttar Pradesh 208016, India. [3]Department of Neurobiology and Howard Hughes Medical Institute, Harvard Medical School, Boston, MA, USA. [4]Mehta Family Center for Engineering in Medicine, Indian Institute of Technology Kanpur, Kanpur, Uttar Pradesh 208016, India. ✉e-mail: guptan@iitk.ac.in

specific odors is less clear[44,45]. It may rely on stereotyped arborization of PNs in the higher brain centers, particularly the lateral horn, which is analogous to the mammalian cortical amygdala[28,46]. An alternative to a combinatorial code is a *labeled line*, in which a volatile molecule with a particular ecological role for a species may be detected by a specific receptor and further processed by a dedicated pathway in the brain, as observed for specific semiochemicals such as pheromones[47–50] or molecules indicating the presence of harmful species[51,52].

It is unclear if the idea of labeled lines can be extended to olfactory behaviors that involve a large number of molecular components—for example, the attraction of a mosquito to a host. The host odor involves hundreds of different types of molecules[53–57]. Moreover, the types and the proportions of components vary considerably between different host species[8,58] and among different members of the same host species[59,60]. These components, even individually, can be detected by the mosquito olfactory system, as confirmed by the measurements of receptor responses[9,10,59,61–67]. The individual components can also generate specific behavioral preferences in mosquitoes[68–72]. While it is possible that a select few of these components or other pheromones in mosquitoes are detected by labeled lines, it seems unlikely that each of these components of the human odor has a dedicated neural pathway for itself in the mosquito brain, which appears similar to the fly brain in size[21,73]. Previous studies of the mosquito AL have used either extracellular recording[74] or calcium imaging with a reporter in all cells or in the receptor neurons[8,13,14,75]; with these approaches, it has not been possible to specifically examine the odor responses of PNs and LNs in mosquitoes.

Here we develop an in vivo preparation for patch-clamp recordings in mosquitoes to target PNs and LNs. Using post hoc dye-filling we examine the glomerular identities and the detailed morphologies of the recorded neurons. We identify different sub-classes of PNs and LNs and compare their morphological and physiological properties. We analyze the odor responses of PNs and LNs and check the representation of the odor identity and the behavioral valence in the PN population. Our results provide a foundation for understanding the olfactory processing in the mosquito brain.

## Results

### Recordings from antennal lobe neurons

We developed an experimental preparation for in vivo recordings in female *Aedes aegypti* by immobilizing the mosquito inside a well in the middle of a custom-built recording chamber in such a way that the dorsal part of the mosquito head was accessible from the top of the well while the body of the mosquito remained below the surface (Fig. 1a; see "Methods"). The perineural sheath from the dorsal and dorsolateral region around the AL was removed gently, taking care to minimize the damage to the tissue. The well was perfused with physiological saline to keep the brain healthy during the experiment. The antennae and the maxillary palps were kept dry and accessible for odor stimulation. To minimize mechanical disturbances during odor delivery, the total flow rate of the final air stream reaching the animal was kept unchanged during the switch between clean air and odorized air (see "Methods"). This preparation allowed us to target individual cell bodies of PNs and LNs and record their spontaneous activities and responses to a panel of odors. One limitation of our preparation was that we could not target cell bodies located in the ventral part of the AL.

### Morphological characterization of antennal lobe neurons

With our experimental preparation, we were able to target cell bodies in the anterodorsal and dorsolateral clusters around the AL. After the recording, we also attempted to fill dye in the recorded neuron followed by brain dissection and imaging for morphological identification. From a total of about 1250 mosquito preparations, we were successful in obtaining recordings along with morphological

identification from 208 PNs and 53 LNs. We found that the distribution of cell bodies around the AL in *Aedes aegypti* is similar to *Drosophila melanogaster*[76–78]: the anterodorsal cluster contains cell bodies of PNs, while the dorsolateral cluster contains cell bodies of both PNs and LNs. By examining the dendritic innervations within the AL, we found that 201 of 208 recorded PNs were uniglomerular, with dense and complete innervation of the corresponding glomerulus (Fig. 1b and Supplementary Fig. 1a); this high proportion of uniglomerular PNs may be because of the targeting of mostly dorsal clusters of cell bodies. Among the remaining multiglomerular PNs, we found five that innervated two glomeruli each, either entirely or partially (Fig. 1b). We also found two PNs in the lateral cluster innervating more than two glomeruli (Fig. 1b). Using the *Aedes* AL atlas provided in ref. 21, we could assign glomerular identity to 175 uniglomerular PNs (see "Methods"). These PNs innervated 40 distinct glomeruli out of 50 described in the atlas.

Next, we checked the axonal projections of PNs in the higher brain centers. We were able to manually trace the axons of 94 PNs (93 uniglomerular covering 36 glomeruli and 1 multiglomerular); the others could not be traced due to the low quality of dye fills or histology. The traces were registered to a female *Aedes* brain template (see "Methods"). Several antennocerebral tracts connecting the AL and protocerebrum have already been described in various insects[79–81]. Among these, the inner antennocerebral tract (iACT), also called the medial antennal lobe tract (mALT), is known to be the most prominent tract. We found that the majority of the PNs sent their axons to the mushroom body calyx and the lateral horn through the iACT (Fig. 1c). PN innervation at the lateral horn was in general more extensive (more branches) than in the calyx, as previously reported in *Drosophila*[81]. We calculated the pairwise neuronal similarity between the lateral horn projections of PNs (Fig. 1d) using NBLAST and then compared the similarity between pairs of homotypic PNs (i.e., PNs innervating the same glomerulus, but not necessarily of the same morphology elsewhere) and pairs of heterotypic PNs. The NBLAST scores for homotypic PN pairs were moderately higher than for heterotypic PN pairs ($0.57 \pm 0.21$ vs $0.42 \pm 0.24$, $P = 4.9 \times 10^{-19}$, two-sided rank-sum test; Fig. 1e), indicating that homotypic PNs have more similar projections in lateral horn. We also observed that PNs innervating dorsomedial glomeruli within the antennal lobe mostly project to dorsal region of the LH, while PNs innervating other glomeruli projected more to the ventral–anterior region in the LH (Fig. 1c).

All the identified LNs in our dataset projected ipsilaterally, although bilateral LNs are known in *D. melanogaster*[27,31]. We analyzed the arborization patterns of 46 out of 53 LNs that had relatively clear stains and grouped them into four morphological sub-classes: *panglomerular* (covering entire AL; $n = 11$), *all-but-few* (covering almost entire AL except for a few glomeruli; $n = 26$), *regional* (innervating a group of connected glomeruli; $n = 7$), and *patchy* (innervating a group of disconnected glomeruli; $n = 2$) (Fig. 2a). The *pan-glomerular* and *all-but-few* sub-classes together can be considered equivalent to the "broad" class of LNs reported in *D. melanogaster*[31]. In most cases of innervation of a glomerulus by an LN, the innervation covered the entire glomerulus, but in some cases, the innervation covered only a region within the glomerulus, suggesting compartmentalization within glomeruli. Next, we tried to check if some glomeruli are innervated more frequently or less frequently by LNs compared to other glomeruli. Given the difficulty in identifying glomeruli and in following weakly stained branches of LNs, we restricted this analysis to 14 landmark glomeruli (I-AD1, I-AD2, I-AM1, I-AM2, I-MD1, I-MD2, I-MD3, I-PC1, I-PD6, I-PL2, I-PL3, I-PL6, I-PM4, and I-V1) that were relatively easy to identify in different brains (Fig. 2b) and 41 LNs whose branches were relatively clear. We found that, compared to other glomeruli, three anterodorsal glomeruli (I-AM1, I-AM2, and I-V1) have relatively fewer innervations by the LNs (Fig. 2c). We note that this result is based on the limited set of 41 LNs, and it is possible that others LNs that were not

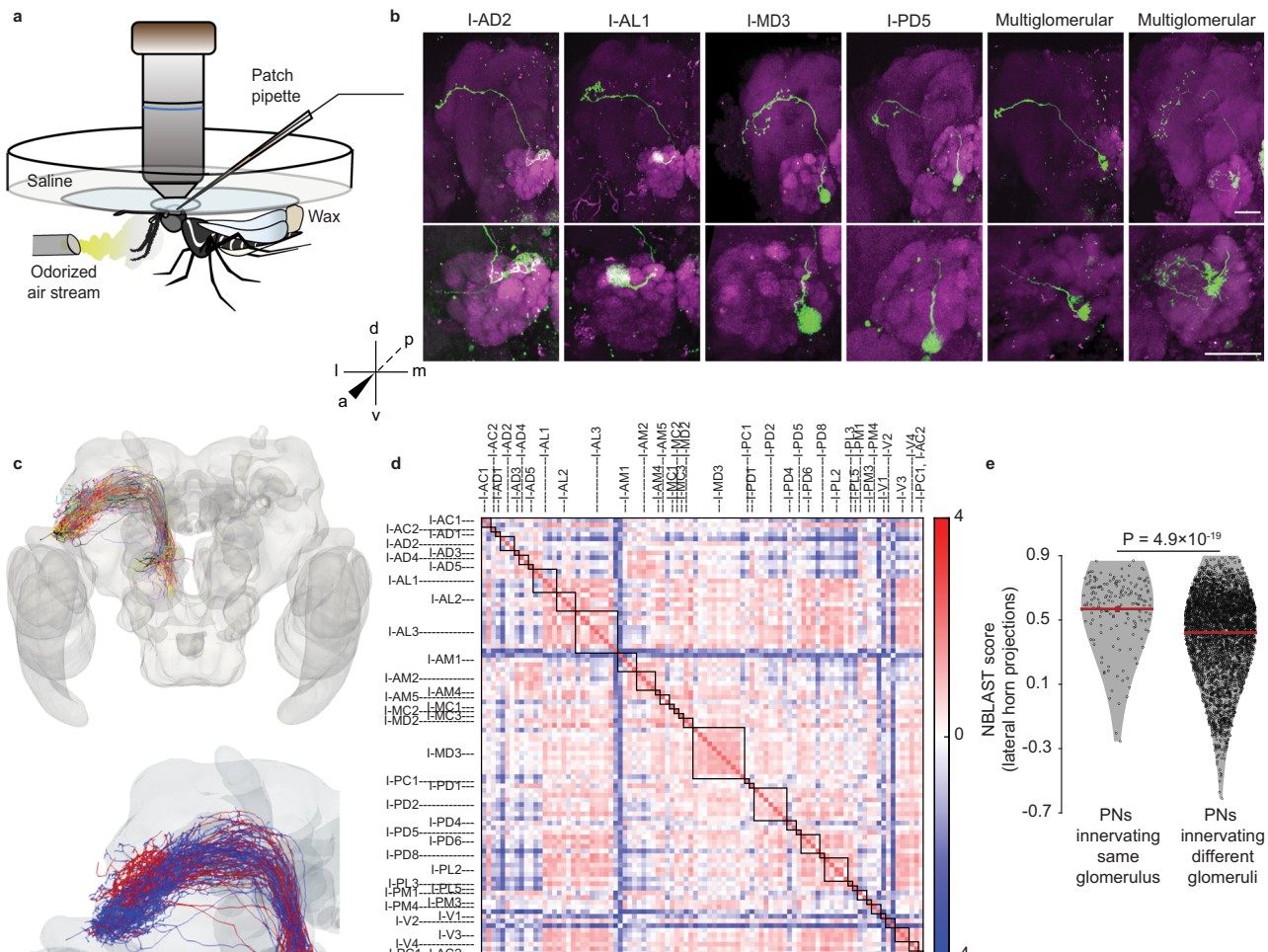

**Fig. 1 | Morphological characterization of projection neurons (PNs) in *A. aegypti*. a** Schematic showing the experimental preparation for in vivo patch-clamp recording. The animal was fixed with glue and wax to a recording chamber placed under the objective lens of an upright, fixed-stage microscope. Odors were delivered to the olfactory organs below the recording chamber and removed using a vacuum tube (not shown) placed behind the animal. Recordings were obtained from cell bodies present in the dorsal and the lateral regions around the antennal lobe (AL) using a glass pipette through a hole in the head cuticle. **b** Examples of some morphological sub-classes of PNs, including 4 uniglomerular (out of 201 obtained), and 2 multiglomerular (out of 7 obtained) PNs. For each PN, the top image shows the complete morphology in a maximum-intensity projection of the image stack (the region displayed includes ipsilateral AL, mushroom body, and protocerebrum); the bottom image shows the dendritic innervations in a maximum-intensity projection from the AL. Co-ordinate axes: dorsal-ventral (d-v),

anterior-posterior (a-p), and medial-lateral (m–l). Green: biocytin or lucifer yellow used to fill the cell; magenta: Dncad (neuropil marker). Scale bars, 50 μm. **c** Top image shows the axonal reconstructions of 94 PNs registered on an adult female *Aedes aegypti* brain template. Each PN sub-class is shown in a different color. The bottom image zooms into the LH projections of PNs innervating dorsomedial glomeruli (red) and other glomeruli (blue). **d** The matrix shows the NBLAST scores (after z-score normalization) indicating the morphological similarity in the proto-cerebral projections of pairs of PNs. PNs are ordered by glomerular identity, marked by the boxes along the diagonal and the labels along the axes. **e** NBLAST scores, indicating morphological similarity of projections to the lateral horn, were higher for pairs of PNs belonging to the same glomerulus (*n* = 171) than for pairs of PNs belonging to different glomeruli (*n* = 4200). *P* value from two-sided signed-rank test is displayed. Red lines: means, error bars: s.e.m. Source data for (**d, e**) are provided as a Source Data file.

sampled in our experiments may have stronger innervations in these glomeruli.

The tracer dye (biocytin) was injected only in the patched neurons. In some samples, however, we found that additional cell bodies, which were not targeted by the electrode, were labeled with the dye (Supplementary Fig. 1b). These sometimes included cell bodies in the ventral region of the AL, which was inaccessible in our preparation, and could not have been accidentally contacted by the electrode. Often the labeling of the additional cell bodies was accompanied by faint signals in multiple glomeruli, suggesting that the labeled neurons are PNs or LNs. Overall, this phenomenon was observed in at least 20 preparations, and points to the presence of gap junctions among the AL neurons in *Aedes aegypti*, as has been previously found in *D. melanogaster*[82]. In most of these cases, the patched neuron was a PN while the additional cell bodies that got filled included PNs or LNs,

suggesting the presence of gap junctions between PN-PN or PN-LN pairs. We do not have sufficient data to comment on LN-LN gap junctions.

## Electrophysiological classification of antennal lobe neurons

Taking advantage of the large dataset of identified PNs and LNs, we asked whether it may be possible to identify a recorded AL neuron as a PN or an LN, in the absence of a dye-fill, simply based on the electro-physiological recordings. We analyzed recordings obtained from morphologically identified PNs and LNs to test this idea. We classified spikes as isolated spikes or bursts (Fig. 3a; see "Methods") and then extracted four electrophysiological features of isolated spikes: spike amplitude, spike half-width, after-hyperpolarization amplitude, and the fraction of isolated spikes (Fig. 3b). These properties differed to dif-ferent degrees between PNs and LNs: on average, PNs had fewer

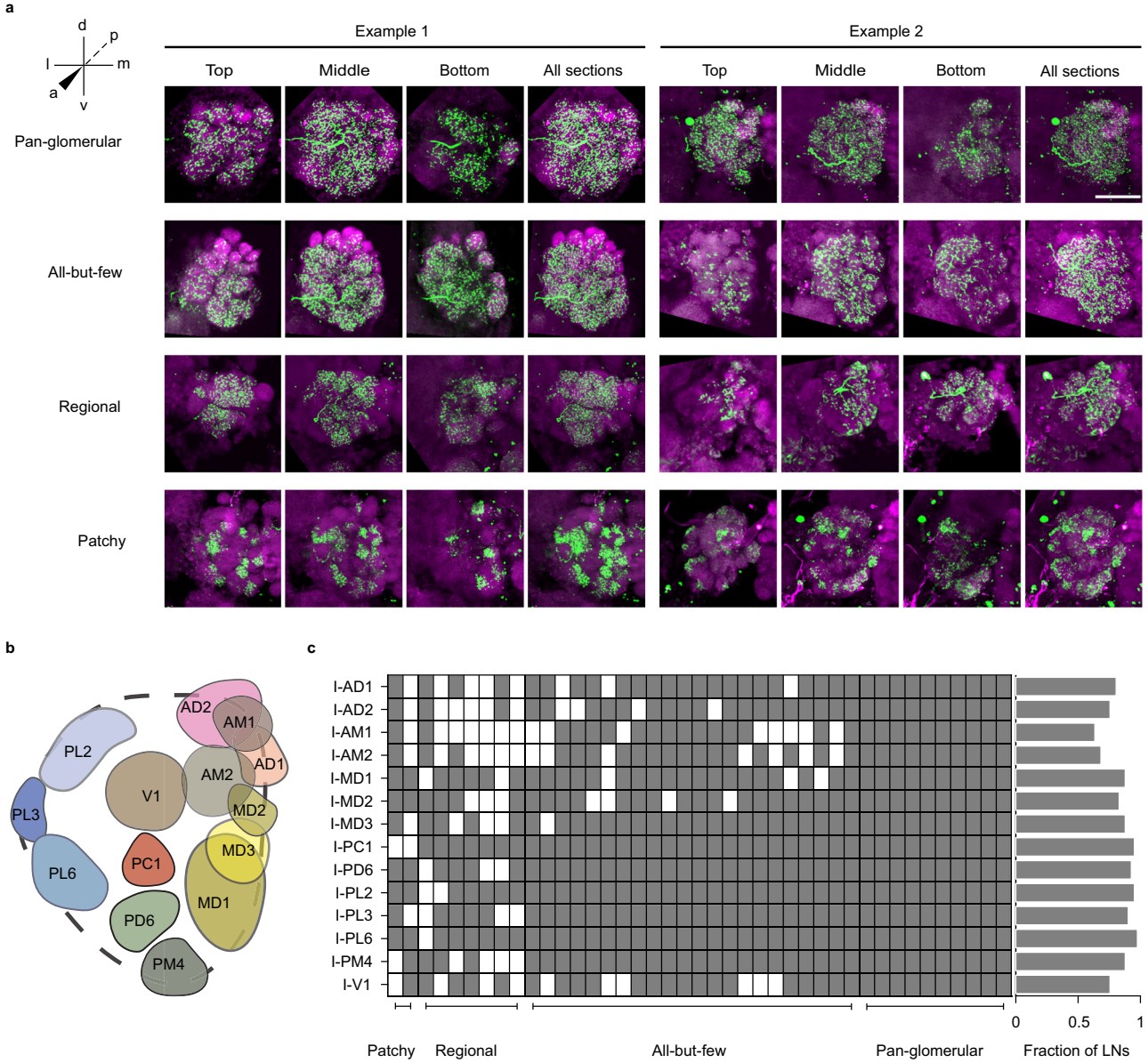

**Fig. 2 | Morphological characterization of local neurons (LNs) in *A. aegypti*.**
**a** Each row shows two examples of LNs belonging to a different morphological sub-class: *pan-glomerular* (*n* = 11 obtained in total), *all-but-few* (*n* = 26), *regional* (*n* = 7), and *patchy* (*n* = 2). The first three images for each example show the maximum-intensity projections from different groups of planes from top to bottom, while the fourth image shows the projection from the entire AL. Green: biocytin/lucifer yellow (cell fill); magenta: Dncad (neuropil marker). Scale bar, 50 μm (same for all images; shown only in the top-right image). **b** Schematic showing the positions of the 14 landmark (easily identifiable) glomeruli scattered across different regions of the AL. **c** Innervations of 41 LNs in the 14 landmark glomeruli (gray: innervation; white: no innervation). LNs are arranged according to the four morphological sub-classes. Right, each horizontal bar indicates the fraction of LNs innervating a glomerulus. Source data are provided as a Source Data file.

isolated spikes and their spikes were smaller, wider, and had less after-hyperpolarization compared to LN spikes (Fig. 3c–f). We focused on these fundamental features of neurons that are independent of odor responses; indeed, the differences observed between LNs and PNs were maintained even when we removed odor-evoked spikes from the analysis (Supplementary Fig. 2a–d). We did not use amplitudes, widths, and after-hyperpolarization amplitudes of burst spikes in this analysis as they were quite variable within a cell; the fraction of burst spikes was implicitly accounted in the fraction of isolated spikes (the two fractions add up to 1). Combining normalized values of the four properties into an electrophysiological feature vector for each neuron, we checked the correlations between pairs of neurons (Fig. 3g). On average, we found positive correlations for pairs of PNs and for pairs of LNs, and negative correlations for PN-LN pairs (Fig. 3h). LNs formed a more

electrophysiologically homogenous group than PNs (within-group correlations of 0.59 ± 0.38 vs 0.23 ± 0.6, $P = 4.5 \times 10^{-63}$, two-sided rank-sum test). An unsupervised hierarchical clustering performed using the four properties was able to group the neurons into two broad clusters, which we labeled as "PN-enriched" and "LN-enriched" clusters as they matched the morphological identification of PNs and LNs with 95% accuracy (Fig. 3i). The high classification accuracy confirms that PNs and LNs have systematic differences in their electrophysiological properties and that it is possible to use these differences to identify a recorded AL neuron as an LN or a PN even without knowing the morphology of the neuron (e.g., in the absence of dye fills). We observed that out of 212 neurons analyzed, only 2 PNs and 6 LNs were placed in the wrong clusters. We looked further into these specific neurons to see why they were misclassified and found that the 2 misclassified PNs

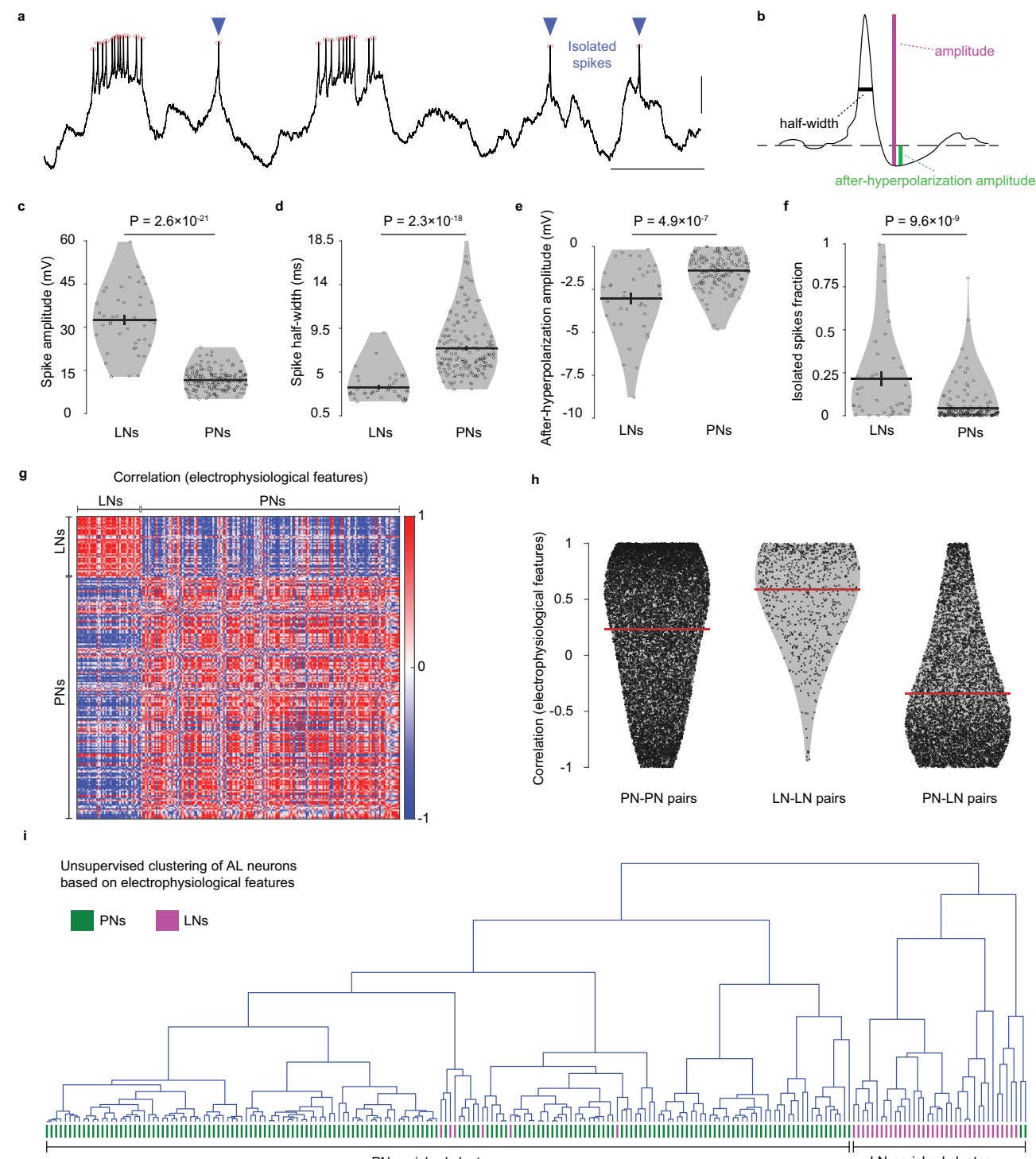

**Fig. 3 | Electrophysiological classification of the antennal lobe (AL) neurons. a** A representative recording trace (3.5 s long) of an AL neuron. Red circles, detected spikes; blue arrowheads, isolated spikes (with no other spikes in ±200 ms); $y$ axis scale bar, 10 mV; $x$ axis scale bar, 500 ms. **b** Illustration of electrophysiological features extracted from the isolated spike. **c**–**f** LNs ($n = 42$) and PNs ($n = 170$) differed significantly in their values for the four electrophysiological features. $P$ values from two-sided rank-sum tests are displayed. Black lines: means, error bars: s.e.m. **g** Correlations in the electrophysiological features between pairs of all AL cells,

arranged according to the morphological class. **h** Correlation values were on average positive for PN-PN ($n = 14,365$) and LN-LN ($n = 861$) pairs, and negative for PN-LN ($n = 7140$) pairs. Red lines: means, error bars: s.e.m. **i** Unsupervised hierarchical clustering of AL neurons using the four electrophysiological features yields two broad classes corresponding to PNs and LNs with high accuracy; the actual identity of each neuron is indicated at the bottom in green (PN) or magenta (LN). Source data for (**c**–**h**) are provided as a Source Data file.

had unusually large fractions of isolated spikes (0.80 and 0.56 respectively, compared to the group mean of 0.04), while the misclassified LNs had unusually large spike half-widths (2 cases), small after-hyperpolarization (1 case), or small spike amplitudes (3 cases).

Further, we checked if the electrophysiological properties depend on the morphological sub-classes within PNs and LNs (physiological sub-classes of local neurons have been previously observed in cockroaches[83]). PNs belonging to the same glomeruli did not show

higher correlations to each other than PNs belonging to different glomeruli ($0.23 \pm 0.60$ vs $0.23 \pm 0.61$, $P = 0.83$, two-sided rank-sum test; Supplementary Fig. 2e), thus showing that the electrophysiological properties do not depend on the glomerular identity of PNs. Within LNs (Supplementary Fig. 2f), we found that neurons belonging to the *pan-glomerular* sub-class had significantly more similar electrophysiological properties than LNs belonging to different sub-classes ($0.81 \pm 0.18$ vs $0.59 \pm 0.38$, $P = 4.7 \times 10^{-4}$, two-sided rank-sum test). Surprisingly, LNs from the *all-but-few* sub-class showed lower correlation to each other than LNs belonging to different sub-classes ($0.49 \pm 0.43$ vs $0.59 \pm 0.38$, $P = 0.02$, two-sided rank-sum test), suggesting that the *all-but-few* sub-class is less homogenous than other sub-classes of LNs.

## Odor responses of projection neurons

PNs are the only neurons that carry olfactory information from the AL to higher brain areas. Therefore, it is important to understand how odors are represented by the PNs. We analyzed the responses of the morphologically identified uniglomerular PNs to a panel of 14 odors (including 7 components of human odor, 3 plant-derived volatiles, 1 carbon dioxide mimic, 1 oviposition attractant, 1 aggregation pheromone, and 1 synthetic repellent (see "Methods"), and solvents (mineral oil and water) (Fig. 4b and Supplementary Fig. 3a); in a few cases, more than one concentration of the odors were tested. In each PN, at least 6 trials were recorded for an odor, although not all odors could be tested with every PN. In a 10 s trial, background activity was recorded for the first two seconds, followed by stimulation with odor for one second. Even in the absence of any stimuli, PNs showed spontaneous firing rates of different magnitudes (mean ± SD: $9.5 \pm 9.3$ Hz). Odors evoked a variety of responses in PNs, as represented by the examples in Fig. 4a. The responses included bouts of excitation (increased firing) and bouts of inhibition (decreased firing). In many cases, the excitation was followed by inhibition (e.g., I-PD6 to DEET .1) or inhibition was followed by excitation (e.g., I-MD3 to 6-methyl-5-hepten-2-one .01). In some cases, temporal patterns of multiple bouts of excitation and inhibition were observed (e.g., I-MD3 to DEET .1 or I-AL3 to propionic acid .01). These temporal patterns were not specific to any odor or any PN but depended on the PN-odor combination. Usually, the response started within 150 ms of the triggering of the odor delivery (this delay includes the time it took for the odor vapors to travel to the animal). However, some odor responses in PNs had longer onset delays of >500 ms, which were consistent across different trials. These delays in the onset of responses cannot be attributed to delays in the delivery of specific odors as we observed that the same odor that generated a delayed response in one PN could generate a fast response in another PN (Supplementary Fig. 3b). A given PN could respond with a small delay for one odor and a large delay for another odor; thus, the onset delays depended on specific PN-odor combinations (Supplementary Fig. 3b, c). The delays in the onset of spiking in a PN and the inhibitory bouts likely reflect odor-specific inhibition received by the PNs through lateral inputs. In some specific cases, prolonged responses to odors were observed that lasted for several seconds after the termination of the odor stimulus (Fig. 4c). Using a statistical criterion to determine if the odor response is significantly different from the background firing (see "Methods"), we estimated the fraction of odors to which each PN responded (mean ± SD = $0.33 \pm 0.25$). These fractions differed for PNs belonging to different glomeruli, although most of them responded to multiple odors (Fig. 4d). These observations of PNs responding to multiple odors with different temporal patterns, including odor-specific onset delays and durations, point to a rich spatiotemporal code for odors.

Next, we evaluated the level of similarity between the odor responses of homotypic PNs. To ensure that the responses are compared using a uniform set of odors, we selected 6 odors that were most frequently tested in our dataset (6-methyl-5-hepten-2-one .01, 1-octen-3-ol .01, dimethyl trisulfide .01, L-lactic acid .1, DEET .1 and

4-methylcyclohexanol .01) and analyzed recordings from 64 PNs (covering 27 distinct glomeruli) in which 6 six odors were tested. We calculated the odor-evoked change in the firing rate for each of the 6 odors, combined these values to create a response vector for each PN, and calculated correlation coefficients between pairs of PN vectors (Supplementary Fig. 4a). We found that the average Pearson correlation for PN pairs innervating the same glomerulus ($0.26 \pm 0.47$) was higher than that of the pairs innervating different glomeruli ($0.08 \pm 0.47$), indicating that odor responses of PNs within a glomerulus are on average more similar ($P = 0.0016$, two-sided rank-sum test; Supplementary Fig. 4b). However, these correlation values had a large spread, indicating that the odor responses of PNs innervating the same glomerulus also varied in many cases. The variation could be because of the differences between differential inputs received by different PNs innervating the same glomerulus (indicating functional diversity among homotypic PNs in mosquitoes) or because of the differences between the individuals from which the PN recordings were obtained. We also considered if errors in the glomerular identification of PNs could explain the observed differences among homotypic PNs but found it to be an unlikely explanation in our dataset: when we restricted the analysis to about half of the homotypic PN pairs using PNs with more confident glomerular identifications, the average correlation for homotypic PN pairs did not increase ($0.19 \pm 0.45$, Supplementary Fig. 4c).

We have seen above that PNs from dorsomedial glomeruli in the AL show a bias towards sending axon projections to dorsal region within LH (Fig. 1c). We asked if these PNs also respond similarly to odors: we compared the odor response correlations for pairs of PNs innervating the dorsomedial glomeruli with correlations for other pairs of PNs. The average correlation for pairs with both PNs innervating dorsomedial glomeruli ($0.21 \pm 0.51$) was higher than for pairs with both PNs innervating non-dorsomedial glomeruli ($0.06 \pm 0.43$, $P = 0.0008$, two-sided rank-sum test) or for pairs with one PN innervating dorsomedial glomerulus and one PN innervating a non-dorsomedial glomerulus ($0.06 \pm 0.45$, $P = 1.1 \times 10^{-7}$, two-sided rank-sum test). Thus, PNs innervating the dorsomedial glomeruli appear to be slightly more similar to each other morphologically and functionally than other PNs (Supplementary Fig. 4d).

## Odor responses of local neurons

Compared to PNs, LNs showed lower spontaneous activity (mean ± SD: $2.6 \pm 2.6$ Hz) Fig. 5a). LNs robustly responded to odors but these responses were in general of lower magnitude than observed in PNs (Fig. 5b). Prolonged excitatory responses were rare in the LN population. In *D. melanogaster*, LNs have been reported to be broadly tuned[27,35]. Here, in *A. aegypti*, we found that LNs varied remarkably in the fraction of odors to which they responded (Fig. 5c). In some cases, LNs showed reliable sub-threshold changes in the membrane potential, suggesting that LNs receive inputs for more odors than they respond to with spikes (Fig. 5d). Next, we checked if responses were similar for LNs belonging to the same morphological sub-classes. We took a dataset of 22 LNs in which six common odors (same as those used for the PN analysis) were tested and estimated similarity as the correlation coefficient between pairs of LN response vectors. We found that odor responses of *pan-glomerular* LNs were significantly more similar to each other than responses of LNs belonging to different morphological sub-classes ($0.55 \pm 0.33$ vs $0.0001 \pm 0.52$, $P = 0.0001$, two-sided rank-sum test; Supplementary Fig. 5a). Thus, *pan-glomerular* LNs are not only more homogenous electrophysiologically (Supplementary Fig. 2f) but are also more homogenous in terms of odors responses compared to other morphological sub-classes, consistent with their broad innervation and higher probability of responding to an odor (Fig. 5c).

As our patch recordings were performed in the whole-cell configuration, the cell body was often removed as the glass electrode was retracted after the recording. But in some cases, where we managed to

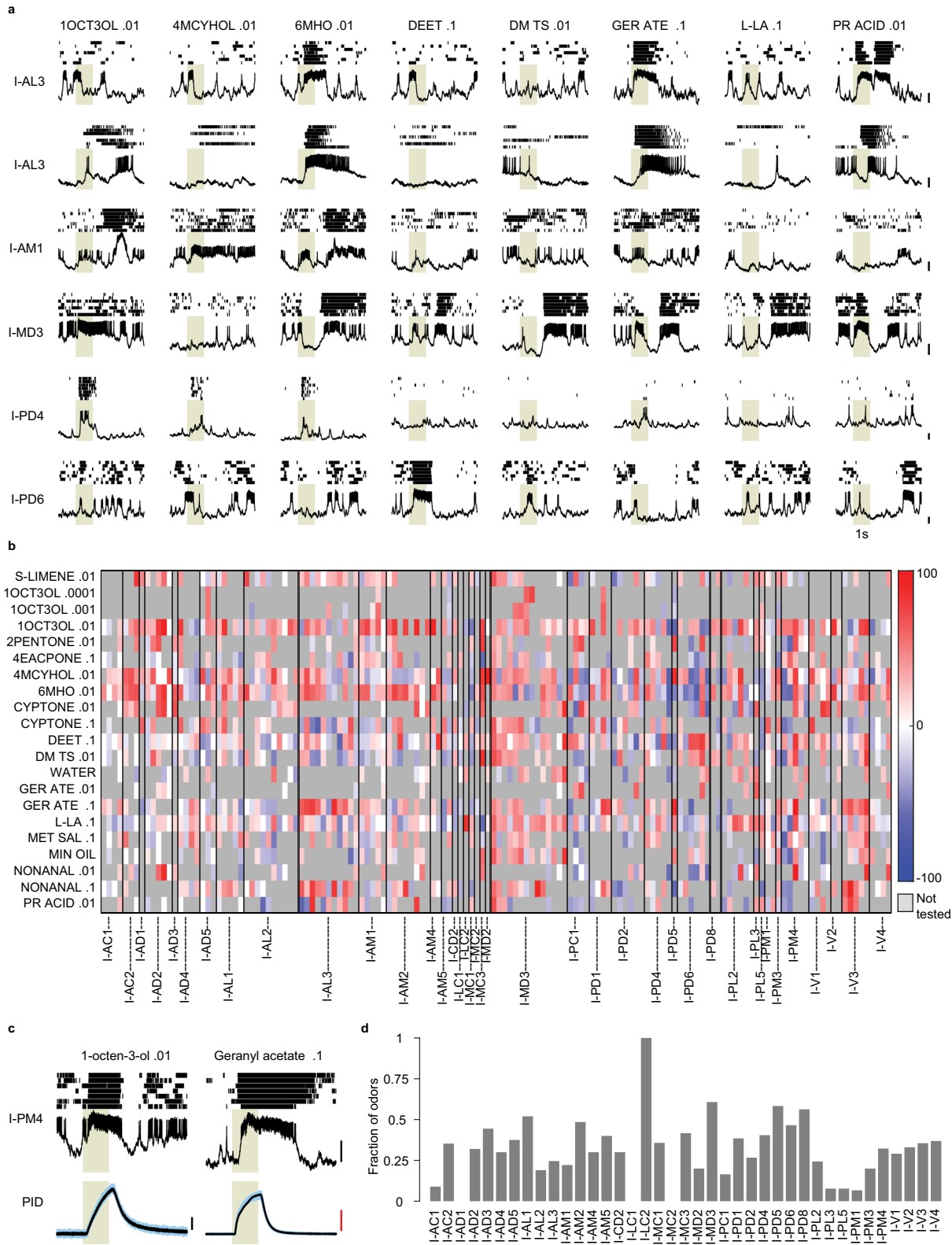

**Fig. 4 | Odor responses of projection neurons (PNs). a** Representative example of odor responses from six PNs and eight odors. Each panel shows a 5-s recording trace along and above it the raster of spikes from seven trials. The first two PNs are homotypic (belonging to I-AL3 glomerulus), recorded from different animals, while the next four PNs belong to different glomeruli. Scale bar, 10 mV. Gray-shaded region indicates the 1-s odor stimulation period. **b** Color-coded change in firing rate (spikes/s) evoked by 14 monomolecular odors (some tested at multiple concentrations) and 2 solvents (mineral oil and water) in 144 PNs, each of which was tested with at least five odors. PNs are arranged by glomerular identity, separated by vertical lines. A gray value indicates that the odor was not tested in the PN. **c** Example of a PN showing a brief response to one odor and a prolonged response to another odor, which continues for 2–3 s beyond the odor stimulation. The bottom panel shows similar delivery profiles for both odors measured with a photo-ionization detector (PID). Scale bars, black: 10 mV; red: 100 mV. **d** The fraction of odors that elicited a response in a PN; values were averaged for PNs belonging to the same glomerulus. Source data for (**b**, **d**) are provided as a Source Data file.

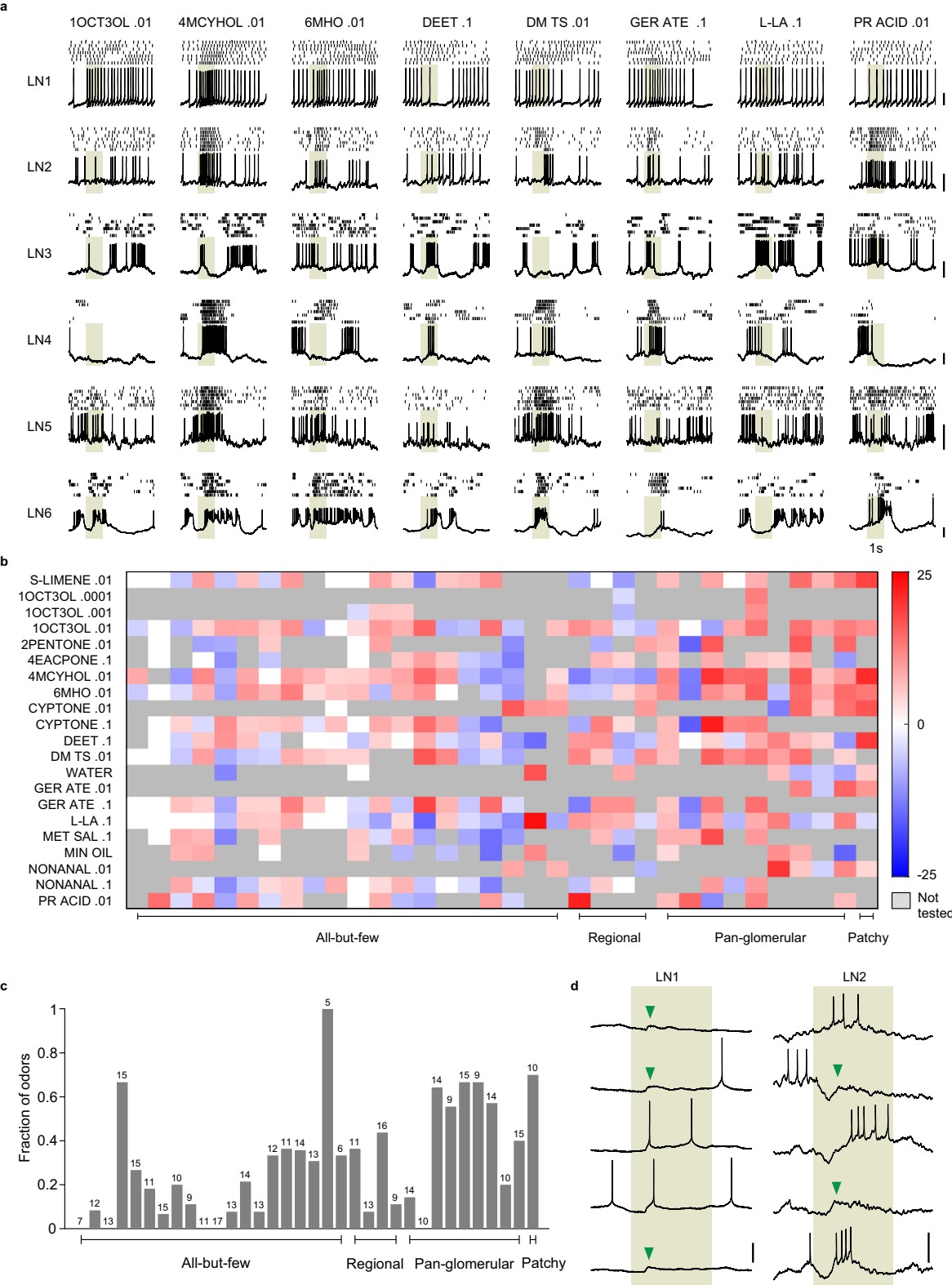

**Fig. 5 | Odor responses of local neurons (LNs). a** Representative example of odor responses from six LNs and eight odors. Each panel shows a 5-s recording trace along and above it the raster of spikes from seven trials. Gray-shaded region indicates the 1-s odor stimulation period. **b** Color-coded change in firing rate (spikes/s) of 34 LNs in response to 14 monomolecular odors (some tested at multiple concentrations) and 2 solvents (mineral oil and water). LNs are arranged by morphological sub-classes. A gray value indicates that the odor was not tested in the LN. **c** The fraction of odors that elicited a response in each LN. The number on the bar

indicates the number of odors tested for that LN. **d** Left: Five trials of recording from an LN with 1-s stimulation of dimethyl trisulfide .01 (shaded region). The LN showed a spike soon after the odor onset in only two of the five trials, but showed an excitatory post-synaptic potential (arrowheads) around the same time in the remaining three trials. Right: Another LN showing similar behavior with 1-s stimulation of cyclopentanone .1. Scale bars, 20 mV. Source data for (**b, c**) are provided as a Source Data file.

cleanly detach the electrode from the cell body, we performed anti-GABA immunostaining to check if the recorded neuron was inhibitory. We found GABA-positive cell bodies in the lateral and ventral clusters around the AL (Fig. 6a). A thick GABA-positive nerve fiber originating from the lateral group and going into the AL was also visible (Fig. 6a). Among the recorded neurons with dye fills and clear GABA stains, we had: six *pan-glomerular* LNs, of which two were GABA-positive and four were GABA-negative; four *all-but-few* LNs, of which two were GABA-positive and two were GABA-negative; and eight PNs with cell bodies in the dorsal cluster, all of which were GABA-negative (Fig. 6b). Presence of GABA-negative LNs suggests that they might be glutamatergic and inhibitory[84] or cholinergic and excitatory[85].

In one experimental preparation in which the antennae were severed to remove the majority of the sensory input (palps were left intact), we succeeded in recording from one I-AL3 PN and one I-PL2 PN simultaneously (Supplementary Fig. 6a). These glomeruli do not receive direct sensory input from the palps[21,86], and expectedly did not respond to odors that are known[87] to activate the sensory neurons in the palp (Supplementary Fig. 6b). But they still showed spontaneous spikes, suggesting indirect inputs from within the antennal lobe (Fig. 6c). Although the spike patterns in the two neurons were not identical, there was a remarkable similarity in these two neurons (Fig. 6c). To test whether this similarity was due to common inputs from other neurons or due to direct or indirect lateral interactions between the two neurons, we performed experiments in which we stimulated one neuron with current injection and observed the effect on the other. We found that while the injection of hyperpolarizing current in either cell had no effect on the other (Supplementary Fig. 7a), stimulation of the I-PL2 PN with depolarizing current resulted in a reliable increase in the firing of I-AL3 PN (Fig. 6d). This effect was directional, as the stimulation of the I-AL3 PN had no effect on the I-PL2 PN. The spike-triggered average of the I-AL3 PN's membrane potential, triggered on I-PL2 PN spikes, suggested the lack of a direct connection between the two neurons (Supplementary Fig. 7b). Together, these results point to an indirect lateral excitation from an I-PL2 PN to an I-AL3 PN, which may possibly be mediated by excitatory LN.

## Combinatorial coding of odors in the antennal lobe

The results so far indicate that a given PN or LN can be activated by multiple odors, and a given odor can activate multiple PNs and LNs (Figs. 4a and 5a). The fraction of PNs activated by an odor depends on the odor identity: some odors like 6-methyl-5-hepten-2-one .01, 1-octen-3-ol .01, and 4-methylcyclohexanol .01 activated about half of the PNs tested, while 4'ethylacetophenone .1 activated 16% of cells, which was the smallest fraction among the odors tested (Fig. 7a and Supplementary Fig. 8). Similarly, each odor activated multiple LNs with the exact number depending on the odor identity (Fig. 7a). Interestingly, over the set of the odors tested, the fraction of PNs and the fraction of LNs activated by an odor were positively correlated ($R = 0.69$, $P = 0.002$, $n = 17$), suggesting that some odors in general elicited more widespread activity than others in the AL (Fig. 7b).

The ability of the higher olfactory centers to distinguish between different odors depends on the inputs they receive from the PN population. We visualized the dynamic odor response of the PN population by reducing its dimensionality to 3 using principal component analysis (see "Methods"). The trajectories followed by different odors in the 3-dimensional space started at the same point but diverged over the course of the 1-s odor stimulation; the trajectories remained separated even after the end of the odor stimulus for another ~1 s and gradually returned to the origin (Fig. 7c). Thus, the odor identity seems to be encoded by the temporally evolving responses of the PN population.

To confirm this, we checked how accurately the odor identity can be inferred from the PN population responses during individual trials. We considered the temporal response vectors (binned into 250 ms

windows) of all PNs for increasing lengths of response periods following the odor onset, and then performed a classification analysis by comparing the population response during a given odor presentation to the templates constructed from other trials of the 6 odors in the dataset (see "Methods"). Compared to 1/6 classification accuracy expected by chance among 6 odors, we found nearly four times higher accuracy (0.67) within 250 ms of the response, which increased to 0.89 in 500 ms, and reached 0.92 in 750 ms, after which the inclusion of additional response periods did not change the performance (Fig. 7d). These results suggest that the PN population encodes the identity of an odor within the first 500–750 ms of the odor response. To understand how many PNs are actually needed in the population for decoding the odor identity, we performed the same analysis by using different numbers of PNs chosen randomly. The results show that the odor identification among the six odors steadily improves as the number of PNs increases and begins to saturate around 40–50 PNs (Fig. 7e, f). Thus, even though individual PNs respond with different levels of promiscuity to odors, large groups of PNs can faithfully encode the odor identity in their temporally patterned responses.

## Innate valences of odors and their representation in the PN activity patterns

We next sought to understand the relationship between the PN population response and the behavioral preference elicited by an odor in female *A. aegypti*. To evaluate innate preferences for the same odor concentrations that are used in electrophysiological recordings, we used a custom-made T-maze olfactometer designed to reveal both attractive and aversive preferences (see "Methods") (Fig. 8a). Briefly, it consisted of a large cuboidal chamber that was divided into two arms: one arm flushed with clean air and the other with the odorized air stream. Mosquitoes were released into the center of the chamber and had to choose between the odorized arm and the control arm. A preference index (PI) for the odor was calculated as the number of mosquitoes in the odorized arm minus the number of mosquitoes in the control arm, divided by the total number of mosquitoes in the two arms. For the same odor dilution, the final concentration reaching the animal was comparable to that in the electrophysiological preparation (Supplementary Fig. 9). We found that 7 odors (4-methylcyclohexanol .01, cyclopentanone .1, 2-pentanone .01, 1-octen-3-ol .01, dimethyl trisulfide .01, 6-methyl-5-hepten-2-one .01, and DEET .1) were significantly aversive (negative PI), one odor (L-lactic acid .1) was significantly attractive (positive PI), and the remaining odors were behaviorally neutral at the tested concentrations (Fig. 8b). At lower concentrations, 1-octen-3-ol and 6-methyl-5-hepten-2-one were found to be attractive (Supplementary Fig. 10a), suggesting that the higher frequency of aversive responses we observed is likely related to the concentrations used and not due to a bias in the assay.

Combining the behavioral and the electrophysiological data, we checked if the odors generating similar behavioral preferences also generated similar activity in PNs. For each pair of odors, we calculated the difference in their PIs and the difference in their PN responses (calculated as 1 minus the correlation between the vectors of odor-evoked responses in PNs that were tested with both odors; see "Methods"). Over all pairs of odors, the difference in PIs was positively correlated with the difference in PN responses ($R = 0.24$, $P = 0.023$; $n = 91$ odor pairs from 14 odors; Fig. 8c). This correlation was specific to PNs; a similar analysis performed with LNs showed no correlation ($R = 0.013$, $P = 0.9$, $n = 90$ odor pairs from 14 odors; Supplementary Fig. 10b). Next, we asked how this behavior–PN correlation depends on the PN population size. To check this, while calculating the difference in PNs responses for a pair of odors, instead of using all the PNs that were tested with the two odors, we randomly subsampled a given number of common PNs, and recalculated the behavior–PN correlations from this reduced PN population; this sampling was performed 20 times to estimate the average behavior–PN correlation for each

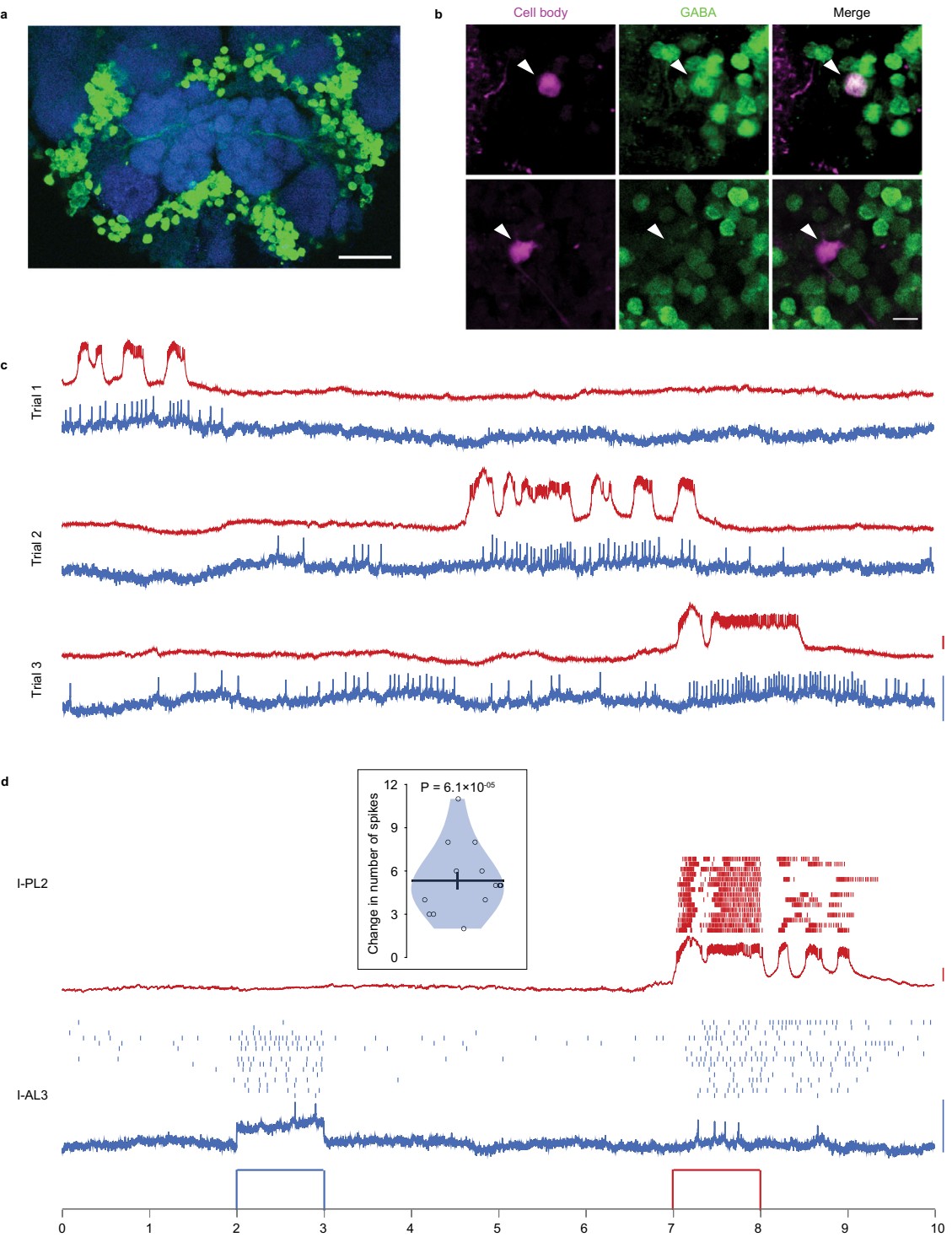

**Fig. 6 | Lateral interactions in the antennal lobe (AL). a** Image showing immunohistology with GABA antibody (green) and Dncad (blue, neuropil marker) around the AL in the *A. aegypti* brain (experiment performed 18 times). GABA-positive cell bodies can be seen in lateral and ventral clusters; a tract of GABA-positive fibers entering the AL from the lateral side can also be seen. Scale bar, 50 μm. **b** Optical sections from two brains (in two rows), in each of which an LN was recorded and filled (magenta) and immunostaining for GABA (green) was performed (experiment performed ten times). One LN (top row) is GABA-positive and the other LN (bottom row) is GABA-negative. Scale bar, 10 μm. **c** Spontaneous activity recorded

simultaneously from an I-AL3 PN (blue) and an I-PL2 PN (red). Three trials of 10 s duration are shown. **d** Depolarizing current of 20 pA was injected during 2–3 s in I-AL3 (blue) and during 7-8 s in I-PL2 (red). Spike rasters for 15 trials (each 10 s long) along with the recording trace of the first trial are shown for both neurons. Inset: the violin plot shows a significant increase in the number of spikes in I-AL3 during 7-8 s period when current is injected in I-PL2 (compared to 6–7 s period) over $n = 15$ trials. *P* value from the two-sided signed-rank test is displayed. Black lines: means, error bars: s.e.m. Scale bars, blue: 5 mV; red: 10 mV. Source data are provided as a Source Data file.

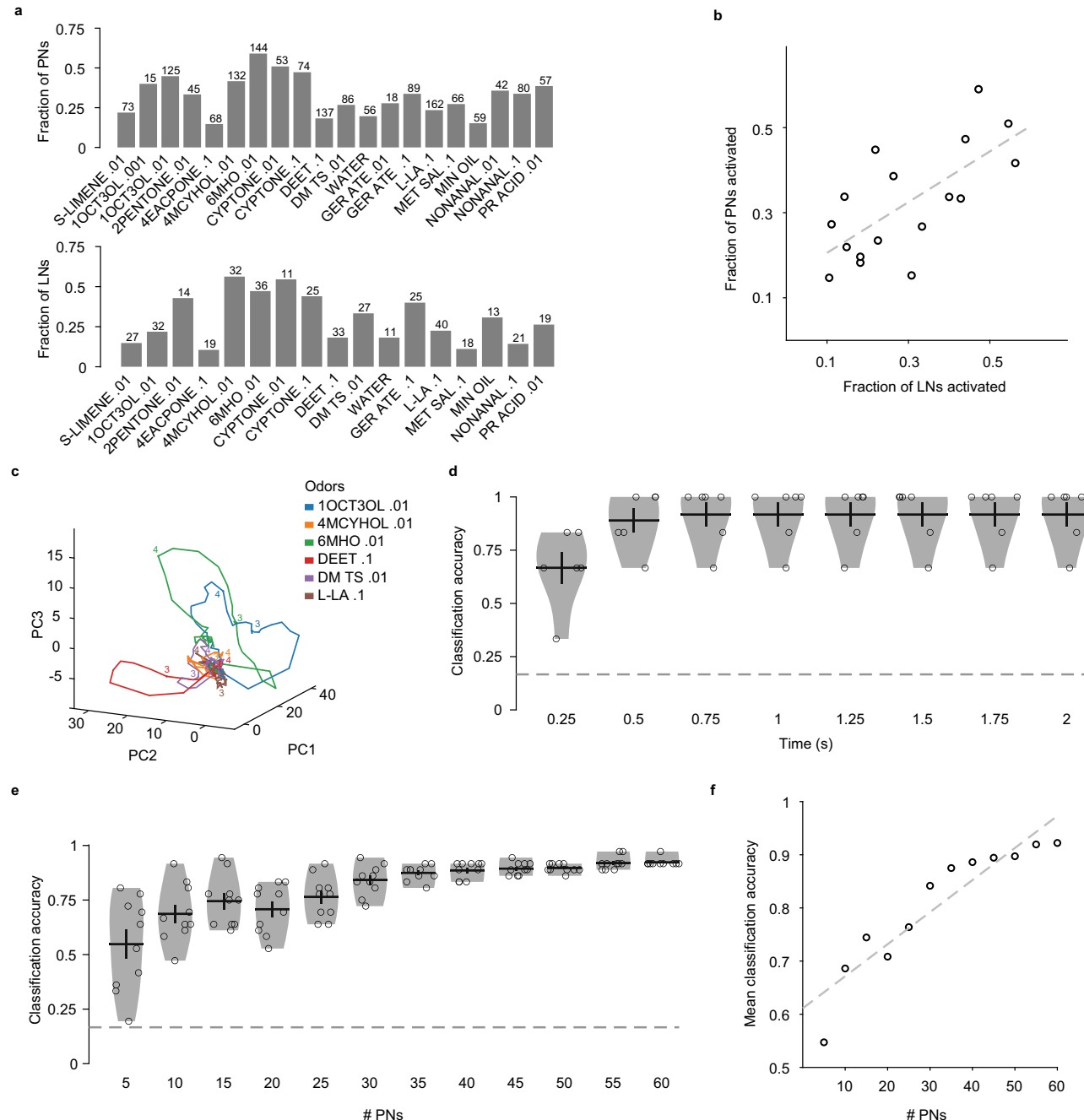

**Fig. 7 | Combinatorial coding of odors in the PN activity. a** Fractions of PNs (top) and LNs (bottom) activated by each odor. The number above each bar indicates the number of cells tested with that odor. **b** The fractions of PNs and LNs activated by odors were correlated (Pearson Correlation $R = 0.69$, $P = 0.002$, $n = 17$ common odors). **c** Trajectories of PN responses to six odors projected on a 3-dimensional principal components (PC) space. The trajectories remained near origin before odor delivery (2 s timepoint) and began to diverge between 2 and 3 s timepoint, and gradually returned to origin (numbers on the plot indicate some timepoints). **d** Accuracy of odor classification based on PN population responses as a function of

the response duration used. Each dot represents the average accuracy for test odors in a given trial. The accuracy for all durations exceeded the chance level (dashed line at 1/6). Black lines: means, error bars: s.e.m. **e** The classification accuracy improves with an increase in the number of PNs taken for the analysis. Each dot represents a different random sampling of the given number of PNs. Black lines: means, error bars: s.e.m. **f** Correlation between mean classification accuracy (calculated in (**e**)) and number of PNs taken (Pearson Correlation $R = 0.93$, $P = 1.5 \times 10^{-05}$, $n = 12$). Source data for (**a**, **b**, **d**–**f**) are provided as a Source Data file.

number of PNs. The results showed that the average behavior–PN correlation increases with the number of PNs considered ($R = 0.95$, $P = 3.2 \times 10^{-24}$; Fig. 8d).

## Discussion

We recorded the responses of LNs and PNs to several odors by adapting the technique of in vivo whole-cell patch-clamp recordings

for the mosquito AL. Multi-electrode recordings and calcium imaging provide population-level information. However, properties of individual neurons in a circuit and network interactions are difficult to assess with these techniques. Intracellular electrophysiology, on the other hand, provides better resolution about how different neurons in a circuit interact, and data can be pooled to get a sense of the final population-level output, especially in a relatively numerically simple

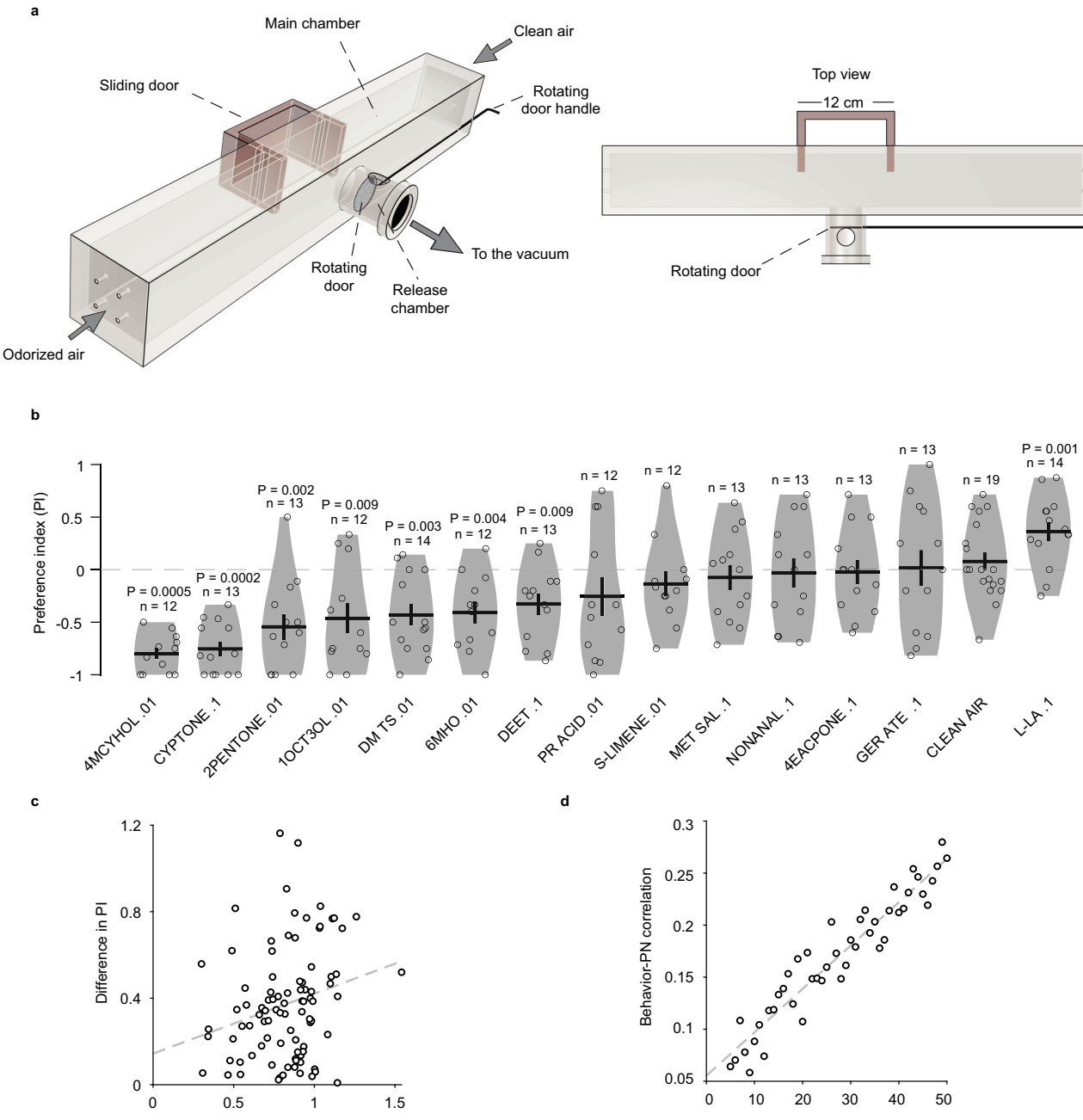

**Fig. 8 | Representation of odor valence in the PN activity. a** Behavioral chamber for evaluating the innate attractiveness and aversiveness of odors in *A. aegypti*. The right panel shows the top view of the chamber. The sliding door (brown) is shown halfway inside the main chamber to indicate how it compartmentalizes the main chamber when it is pushed inside (it is fully open during the experiment and pushed fully inside at the end of the experiment). Female mosquitoes were placed in the release chamber 30 min before the experiment. At the start of the experiment, a rotating door separating the release chamber and main chamber was opened. One side of the main chamber was flushed with odorized air and the other side with clean air. After 5 min, the sliding double door was closed, and the mosquitoes on each side were counted to calculate a preference index. **b** Preference index (PI) for 14 monomolecular odors. *P* values are calculated from two-sided signed-rank test comparing PI to 0; insignificant *P* values are not displayed; *n* (number of repeated experiments) is indicated above each plot. Black lines: means, error bars: s.e.m. **c** Difference in preference indices (PI) is positively correlated with the difference in PN responses for odor pairs (Pearson Correlation $R = 0.24$, $P = 0.02$, $n = 91$ odor pairs). **d** The behavior–PN correlation increases with the number of PNs used in the analysis (Pearson Correlation $R = 0.9$, $P = 3.2 \times 10^{-24}$, $n = 46$). Source data for (**b**–**d**) are provided as a Source Data file.

nervous system such as that of an insect. Another advantage of using this technique is the ability to perform dye fills and post hoc histology that reveals the morphology and therefore the identity of the recorded neuron. This is particularly useful in systems where genetic labeling is not easy, such as with mosquitoes. However, one limitation of in vivo intracellular electrophysiology for small neurons is that it requires a very delicate preparation (including removal of a part of the cuticle and perineural sheath above the brain), and the slightest vibration can derail the experiment; to record about 250 cells, we had to go through nearly 1250 mosquito preparations. The practical challenges also limit the recording time and thus the number of odorants tested per cell. And because we wanted to test each cell with different types of odorants, many of the odorants could be tested at a single concentration only. Thus, for assessing dose-response

relationships and the neural representations at a larger range of dilutions, future studies that test multiple concentrations of specific odorants will be required. In the current study, we performed experiments on mated but non-blood-fed female mosquitoes, which need blood from hosts. As the olfactory responses depend on the internal state of the animal[88,89], we expect that odor responses of PNs and LNs might change for unmated or blood-fed females and for males.

We found that the morphological organization of AL neurons in mosquitoes is similar to that of *Drosophila*, but with some differences. Unlike in *Drosophila* where more than half of the PNs are multiglomerular[28], 201 of the 208 PNs in *A. aegypti* we observed were uniglomerular; however, the low number of multiglomerular PNs in our dataset may be related to the fact that cell bodies in ventral AL were not accessible in our preparation. Further work targeting cell bodies in all clusters around the AL is needed to know the exact proportion of multiglomerular PNs in mosquitoes. We found many LNs with broad innervations (*pan-glomerular* and *all-but-few* sub-classes) and some LNs with selective innervations in glomeruli (*regional* and *patchy* sub-classes). Similar sub-classes have been described in *Drosophila*[27,31], with one notable difference: we did not observe any bilateral LN in our sampling from mosquitoes. Taking advantage of the detailed information available in intracellular recordings, we were able to extract four features of spikes that could predict whether a recorded neuron is a PN or an LN. This result is consistent with studies on other insects showing differences between PNs and LNs[35,90–92], and will help in identifying neurons in cases of unsuccessful dye-filling or damage to the brain tissue while handling.

We also measured the behavioral preferences for the odors at the same concentrations as used in our electrophysiology experiments. The behavioral assays were performed using a custom T-maze olfactometer with a small stem (a small cylindrical release chamber). A long stem can be problematic with aversive odors as the animals might not come out of the release chamber to fly upwind and reach the decision point in the odor arm or the control arm. In our olfactometer, a small release chamber resulted in mosquitoes entering the decision-making zone more frequently. The decision points within the two arms were defined by the sliding doors, which were 6 cm inside the arms from the center of the chamber (pushing them further towards the ends of the arms would reduce the number of responding mosquitoes and make it harder to detect aversion reliably). We used active airflow to precisely control the concentration of odor within the chamber. Our large dataset of neuronal responses, along with behavioral data, allowed us to explore the general principles of odor representation in the AL. We found that although an individual PN typically responds to multiple odors, the pattern of activity in the PN population is unique for each odor. These patterns for different odors are separated well enough to allow reliable odor discrimination. We also found that the odors become more discriminable when responses from more PNs are considered. Odor pairs that elicit similar activity in the PN population elicit similar behavioral preferences in mosquitoes. Together, these results support a combinatorial code for odors in the PN population.

Many of the odors used in this study have been at the center of mosquito olfaction research. For example, 6-methyl-5-hepten-2-one (6MHO, also known as sulcatone) is an important skin component and evolution of *A. aegypti* to bite humans has been attributed to increased expression of receptors for 6-methyl-5-hepten-2-one[2]. However, a high concentration of sulcatone in human odor has been associated with reduced attraction[68]. In our behavioral experiments, 6-methyl-5-hepten-2-one .001 was attractive but 6-methyl-5-hepten-2-one .01 was aversive and in our patch recordings, we found that 6-methyl-5-hepten-2-one .01 activates a large subset of PNs (Fig. 7a). DEET is another odor that has been of great interest as the most widely used repellent against mosquitoes. Despite being in use for a very long time, the mode of action of DEET has been puzzling and appears to be species-specific[93–95]. Using calcium imaging, Lahondère et al., 2020 recently showed that DEET activates AM2 glomerulus in *A. aegypti;* however, since the calcium reporter used in this study was driven by the ubiquitin promoter, it was not possible to pin down this activity to a specific cell-type within the glomerulus. With our recordings, we found that PNs innervating I-AM2, I-AM4, I-MD3, I-PD6, and I-PM4 respond to DEET (Fig. 4b). In our behavioral experiments, mosquitoes were repelled by DEET vapors mixed with the background air stream. Together, our results show that DEET is detected as an odor in *A. aegypti* and activates multiple PNs in the antennal lobe. Previous studies have also reported additional mechanisms through which DEET functions in *A. aegypti*, such as interactions with host odors[96–98]. The PN-odorant combinations tested in our study, along with the information about the cell types and putative interactions in the mosquito AL, provide a foundation for understanding how a large number of behaviorally relevant odorants are processed in the mosquito brain.

## Methods

### Animal stock

All experiments were performed with *Aedes aegypti* (Linnaeus) Liverpool strain. Larvae were reared at $28 \pm 1\,°C$ and $60 \pm 10\%$ relative humidity in 500 ml round plastic containers containing deionized water and fed on powdered fish food (TetraBits). Pupal cups containing 150–200 pupae were placed in a mesh cage (L:30 × B:30 × H:30 cm) for adults to emerge. Adults were fed on a 10% sucrose solution using a cotton feeder and kept at $25 \pm 5\,°C$ and $60 \pm 15\%$ relative humidity under a photoperiod of 14 hr:10 h (light:dark). Females were blood-fed using mice (*Mus musculus BALB/c*) kept in the same environmental conditions, and eggs were collected in plastic containers lined with filter paper strips. For experiments, non-blood-fed mated female mosquitoes of age 4–8 days were used. Mice handling was done as per the approved protocol of the Institute Animal Ethics Committee of the Indian Institute of Technology Kanpur.

### Animal preparation and electrophysiology

In vivo whole-cell patch-clamp recordings were obtained from cell bodies around the AL in *Aedes aegypti*. The mosquito was anesthetized on ice and the legs were immobilized using wax. All subsequent steps were performed at room temperature. The animal was then transferred to an aluminum foil covering a small hole in a plastic cup. The foil above the hole was cut and adjusted to hold the mosquito in such a way that the head and the thorax of the mosquito were accessible from the top while the remaining part was hanging below. Wings and the proboscis were immobilized using wax and the proximal segment of the antennae was pasted to the anterior part of the head using an epoxy adhesive (Araldite). A piece of thin plastic wrap with a small rectangular cut in the center was pasted on the mosquito above the head such that the dorsal part of the head remained exposed. The animal was then transferred from the plastic cup to a hole in a recording chamber, made of a petridish, and the sides of the plastic wrap were sealed onto the surface of the chamber. This way the dorsal part of the head was accessible from the top while the olfactory organs were below the plastic sheet. This ensured that the olfactory organs remained dry and exposed to odor stimulation during experiments. The well was then filled with saline and a window was cut in the exposed part of the head using a sharp tungsten needle. The tracheae were removed, and the perineural sheath was partially removed to expose the cell bodies. In most of the samples, a brief exposure to 0.5% collagenase type IV was used to facilitate de-sheathing (this did not cause any noticeable difference in the recorded activity of the cells). The brain was perfused continuously with saline bubbled with 95% $O_2$/5% $CO_2$. The saline composition was as follows (in mM): TES (12.5), Glucose (8), Sucrose (2), NaCl (120.5), Trehalose dihydrate

(6.25), $NaHCO_3$ (17), $NaH_2PO_4$ monohydrate (0.6), KCl (3.15), $CaCl_2$ dihydrate (1.6), $MgCl_2$ hexahydrate (2.9). The pH was adjusted to 7.3 and osmolarity was set to 300-305 mOsm. The animal preparation was placed under Nikon FN-1 microscope with 40× water immersion objective. Patch-clamp electrodes (6.5–9 MΩ) were pulled from glass capillaries (1.5 × 0.86 mm) using electrode puller (Sutter Instrument P1000). Recording electrodes were filled with an internal solution composed of the following (in mM): Potassium aspartate (140), HEPES (10), MgATP (4), $Na_3GTP$ (0.5), EGTA (1.1), KCl (1), KOH (10). The pH of the internal solution was adjusted to 7.1 and osmolarity was set to 295 mOsm. Biocytin (0.5%) or Lucifer Yellow (0.1%) was added to the internal solution.

Recordings were acquired in current-clamp mode using a Multiclamp 700B amplifier (Molecular Devices), sampled at 20 kHz and low-pass filtered at 10 kHz using Pclamp10 software (Molecular Devices). A small constant current (of magnitude less than 30 pA) was sometimes injected after breaking in, to bring the membrane potential between −45 and −60 mV. Within an experiment, seven trials were recorded for each odor. The total duration of one trial was usually 10 s and odors were delivered for 1 s after 2 s of trial onset. However, if a cell showed prolonged odor response, the duration of the trial was increased to make sure that the firing rate returns to baseline before the onset of the next pulse of odor. Tracer dye (0.5% Biocytin or 0.1% Lucifer Yellow) was iontophoretically injected into the cells from which stable recordings were obtained. To fill the dye, a hyperpolarizing pulse (500 ms) of 3 nA was applied at 1 Hz for 20–40 min after recording. In paired recordings, one cell was filled with biocytin, and the other cell was filled with lucifer yellow.

### Histology

After the recordings, the brains were dissected out and fixed in 4% PFA at room temperature for 30-60 mins. Quick washes with PBS were followed by incubation in PBS containing 0.2% Triton-X (PBS-T) for 1 h. Then, the brains were incubated in 5% normal goat serum at room temperature for 2–3 h or kept at 4 °C for overnight incubation. After this, the samples were incubated in 1:30 rat anti-DN-cadherin (DN-EX #8, Developmental Studies Hybridoma Bank) and 1:500 rabbit anti-GABA (A2052, Sigma-Aldrich) or 1:200 rabbit anti-Lucifer yellow (A5750, Molecular Probes) for 1–2 days at 4 °C. Brains were washed in PBS-T for several hours at room temperature and incubated with 1:500 goat anti-rat with Alexa 405 (ab175671, Abcam), 1:500 goat anti-rabbit with Alexa 633 (A21070, Molecular Probes) or goat anti-rabbit with Alexa 488 (A11008, Molecular Probes) and 1:10^5 Streptavidin with Alexa 488 (S11223, Molecular Probes) or Streptavidin with Alexa 568 (S11226, Molecular Probes) for 1–2 days at 4 °C. Then the brains were washed multiple times with PBS and PBS-T for 3–4 h and mounted in Vectashield anti-fade mounting media (H-1000, Vector Laboratories). Morphological images were obtained using Nikon A1R MP+ microscope and NIS-C software version 5.1 (Nikon).

### Identification of glomeruli

To identify the glomeruli, we compared the histological images with the atlases provided in refs. 21 and [73]. Our images matched the Ignell et al., 2005 atlas more closely and hence we have assigned the glomerular identity using this atlas; we have added a prefix "I-" to glomerular names to indicate this. Some glomeruli are comparatively large and invariant in shape and location and could be identified easily in different samples; these served as landmark glomeruli. Some glomeruli were challenging to identify due to unclear boundaries or variations in shape, size, or location. To track and improve our confidence in the identification, two members from our team independently labeled each histological image and assigned a confidence score (0–5). If their initial identifications for an image differed, the two members reanalyzed the image together to arrive at a consensus. PNs with a confidence score ≥2 were analyzed. The following

glomeruli were included in the group of dorsomedial glomeruli: AM1, AM2, AM3, AM4, AM5, AL1, PM1, AL2, AD1, AD2, AD3, AD4, AD5, AC1, MD2, PD1, AC2.

### Behavioral experiments

For behavioral experiments, a custom-built acrylic behavioral chamber was used. It consisted of a main cuboidal chamber (L:60 × B:9 × H:9 cm) and a cylindrical release chamber (L:8 × D:5 cm) that was attached to the center of the rear panel of the main chamber. A rotating door at the junction separated the release chamber from the main chamber. The other end of the release chamber had a copper screen. A sliding double door was placed in the middle of the main chamber to divide it into two arms on the side and a small middle region between the two doors that connected to the release chamber. Airstreams (5 L/min) entered from the sides of the chamber through four tubes attached at each side panel. A vacuum tube was placed behind the copper screen of the release chamber to maintain the air-flow entering from the sides and exiting at the middle. Odorized air came in from one side and clean air (control) from the other. LED lights were placed at both ends of the main chamber to illuminate both arms equally. During the experiment, a white cardboard cover was placed over the behavioral chamber to avoid visual biases. A black stripe pattern was put on all the sides and bottom of the chamber to provide visual features that might help mosquitoes to navigate. The olfactometer was kept in another acrylic chamber (L:120 × B:60 × H:60 cm) to isolate it from the surroundings.

For each experiment, 18–20 female mosquitoes were taken from rearing cages and transferred to a starvation cage for 24 h where they had access to water but not sucrose. Just before the experiments, the mosquitoes were moved to the release chamber and allowed to acclimatize for 30 min. At the start of the experiment, the odorized stream was switched on and the odor was delivered continuously in pulses of 1 s with gaps of 4 s. After 30 s, the rotating door of the release chamber was opened so that the mosquitoes could fly into the main chamber. Over the next 5 min, mosquitoes were allowed to move free in the main chamber. At the end of the experiment, the sliding door was closed and mosquitoes in the two side arms were counted and considered as responding. The mosquitoes remaining in the release chamber or in the middle part of the main chamber (between the two sliding doors) were also counted and considered to be non-responding. Only those experiments where the number of responding mosquitoes was ≥5 were considered for further analyses. Preference index for the odor was calculated as $(N_o - N_c)/(N_o + N_c)$, where $N_o$ and $N_c$ are the numbers of mosquitoes in the odorized arm and the control arm, respectively.

After each experiment, the chamber was cleaned with 70% (v/v) ethanol and flushed with clean air. The side of the odorized air was randomly assigned in each experiment to minimize the bias from any unplanned difference between the left side and the right side of the chamber. The temperature inside the chamber was maintained between 25 and 30 °C and the relative humidity was between 45 and 80%. Any experiment in which the temperature or the humidity inside the chamber was out of this range or the number of responding mosquitoes was less than 5 was excluded from the analysis. For each odor, data from 12 to 14 experiments were obtained.

### Odor delivery

Fourteen odorants were used in this study including seven components of human odor: 1-octen-3-ol (1OCT3OL), 2-pentanone (2PENTONE), 6-methyl-5-hepten-2-one (6MHO), dimethyl trisulfide (DM TS), L-lactic acid (L-LA), nonanal (NONANAL), and propionic acid (PR ACID); three plant-derived molecules: (S)-limonene (S-LIMENE), geranyl acetate (GER ATE), and methyl salicylate (MET SAL); one oviposition attractant[99]: 4-methylcyclohexanol (4MCYHOL); one mimicking carbon dioxide[66]: cyclopentanone (CYPTONE); one aggregation

pheromone: 4'ethylacetophenone (4EACPONE); and one synthetic repellent: N,N-diethyl-meta-toluamide (DEET) (Supplementary Table 1). Some of the odorants were tested at more than one concentration. Dilutions (v/v) were done with mineral oil except for L-lactic acid, which was diluted in water. In all, 2 ml of the odorant was placed in a 50-ml glass bottle. For electrophysiology, a stream of dehumidified and filtered compressed air (2 L/min) was directed at the animal throughout the recording. The 2 L/min flow included a constant stream of 1.8 L/min and a flexible stream of 200 ml/min that passed either through a clean empty vial (during background period) or through the saturated headspace of an odor vial (during odor stimulation), controlled by a three-way distributor solenoid valve (Product code: 11-13-3-BV-24F88, Parker Hannifin), ensuring that there was no change in the total airflow during odor stimulation. In this arrangement, the odorized stream got diluted by a factor of 0.1 after mixing with constant stream; the dilution indicated against each odor throughout this manuscript is the final dilution coming out of the outlet tube (further dilution caused by mixing of the outlet stream with the ambient air around the animal is not taken into consideration). The outlet was kept 8–10 mm away from the animal. The delivery of odor was confirmed regularly using a photo-ionization detector (200B miniPID, Aurora Scientific). For the behavioral experiments, a constant stream (4.5 L/min) of humidified filtered air was directed into the behavioral chamber from each end of the main chamber. For odor stimulation, an air stream of 0.5 L/min was odorized by passing through an odor vial and added to one side of the chamber, while an equivalent stream of clean air was added on the opposite side. In all the setups, odor tubes were replaced frequently to minimize any contamination and odor vials were replaced periodically to ensure stable odorant concentration.

## Morphological analysis

**Image registration and neuronal similarity analysis.** In order for neuron morphologies acquired from separate brains to be compared, light microscopy images of brains must be registered to a standard template brain space[28]. For ease of analysis, these neurons can then be "skeletonized" and exported as SWC files. We aimed to register our neurons into an open-source female *Aedes aegypti* brain template (obtained from mosquitobrains.org, courtesy of Meg Younger and Leslie Vosshall). Standard practice in the field is to use the Computational Morphometry ToolKit (CMTK)[100] to register brains with a neuropil stain, to a template. However, due to damage to our brains during the recording process, this proved inaccurate. Instead, we performed a landmarks-based registration, marking cross-mapped control points (e.g., specific AL glomeruli, specific tract bends, etc.) between each sample brain and our reference brain, up to 25 control points per brain hemisphere. We used a thin plate spline registration[101] implemented in the R package *Morpho*[102] to create a warping deformation that registered each sample brain to the template, and could be used, via *Morpho* and the *natverse*, to register each of our neuronal reconstructions to the template. Our neuronal reconstructions were of the axonal projections of PNs, and were manually traced using Neutube (Feng et al., 2015)[103] and reconstructed using the SNT plug-in ImageJ. A 3D brain mesh was obtained from the insectbrainDB.org API[104] using the R package *insectbrainr*[28]. Reconstructed neurons were analyzed in R using the *natverse*[28]. The similarities of the arborizations of the registered PN traces in the protocerebrum neuropil were calculated using NBLAST[105]. In order to control for overall neuron size, we resampled neurons to a 0.5 micron spacing for each point in the skeleton, and used normalized NBLAST scores (raw NBLAST score for neuron A to B comparison, divided by the score or a self match, i.e., A to A). In order to compare only the morphology of neurons' axons outside of the mushroom body, we removed all cables from each neuron prior to the most lateral axonal arborization. To remove the influence of cofasciculating tracts on our NBLAST result we also pruned

away the higher strahler order cable in these isolated axons, i.e., the central tract, leaving only the axonal branches.

**Glomerular innervation map of LNs.** Since the shape, size and location of glomeruli vary from one sample to another, we focused on 14 landmark glomeruli (I-AD1, I-AD2, I-AM1, I-AM2, I-MD1, I-MD2, I-MD3, I-PC1, I-PD6, I-PL2, I-PL3, I-PL6, I-PM4, and I-V1) whose identity could be confirmed reliably based on one or more of the following features: a specific location, a unique shape, a noticeably larger or smaller size compared to neighboring glomeruli. We manually examined LN arborizations in each of these glomeruli in the image stacks.

## Electrophysiological feature analysis

Spikes were detected from low-pass-filtered voltage traces using custom code in MATLAB 2020b. Spikes were classified into bursts or isolated spikes: a spike was considered isolated if there were no other spikes within 200 ms on either side of it. Next, for each cell, we extracted the following features from the entire duration of the recordings:

I. *Spike amplitude:* For isolated spikes, the distance between the peak of the spike and the left trough or the right trough (whichever gives a larger amplitude) was taken as the amplitude of the spike. For bursts, the amplitude was calculated in the same way for the first and the last spikes of the burst and mean of the two was used. If a cell had isolated spikes, the mean amplitude of isolated spikes was used for further analysis, otherwise the average amplitude from the bursts was used.

II. *Spike half-width:* The width of the spike at half the height was calculated and averaged over all isolated spikes.

III. *After-hyperpolarization amplitude:* In band-pass filtered traces (5–500 Hz), we calculated the lowest value of the membrane potential within 10 ms after a spike and from it subtracted the average value of the membrane potential in the window of 20–70 ms after the spike. This value was usually negative and was averaged over all isolated spikes; occasional positive values (due to unexpected fluctuations in the membrane potential) were ignored.

IV. *Isolated spikes fraction:* The number of isolated spikes was divided by the total number of spikes in a cell.

We generated a feature matrix with individual cells as rows and mean values of the above features as columns. Each feature was normalized such that when all the cells are considered the mean of each feature is 0 and the standard deviation is 1. In about 5% of the cases, the feature matrix had missing values (e.g., when a cell did not have any isolated spikes). To enable hierarchical clustering, we interpolated the missing values in the following manner. We made a reference set of cells for which all features were available. For a cell with one or more missing values, we used the available features of that cell to calculate its similarity to each cell in the reference set. Then we estimated a missing feature of that cell as the weighted average of the corresponding values from the reference set, with similarity values as weights.

To include reliable data for this analysis, we manually assigned a quality index on a scale of 0–5 for each cell, based on the resting membrane potential after break-in, background noise (as compared to other cells recorded on the same day), average spike size, and the amount of current that had to be injected to stabilize the cell. Cells with quality index greater than 3 were used for this analysis; this corresponded to >80% of morphologically identified neurons (170 of 208 PNs and 42 of 53 LNs).

## Analysis of odor responses

For each cell-odor pair, the response intensity was calculated by subtracting the background firing rate from the firing rate during the

response duration for each trial and was then averaged over all trials available for that pair. In each trial the odor valve was turned on at 2 s timepoint; the first 2 s were used as the background duration and the 2.1–4.1 s interval was used as the response duration (considering 100–150 ms gap between the switching on of the odor valve and the odor actually reaching the animal). Although the odor was delivered for 1 s, we used a 2-s response duration because the odor response often lasted longer than the odor stimulus. Cells that did not show any spike were excluded from this analysis.

For estimating the fractions of cells that responded to an odor or the fraction of odors that activated a given cell, we classified each cell-odor pair as "responding" or "non-responding" using a statistical criterion. We divided the response duration into two 1-s bins and compared the spiking rate in each bin over all trials to the spiking rate in the background period using a two-sided signed-rank test. If the test showed a significantly high or low firing rate in either bin compared to the background ($P < 0.05$), the cell-odor pair was labeled as "responding". The fraction of odors to which a cell responds was determined for only those cells in which at least five odors were tested. For PNs, we took the average over all homotypic PNs for each glomerulus. For LNs, we calculated the fractions for individual cells.

### Visualization of the dynamic PN population response as odor trajectories

This analysis was performed using the dataset of 6 common odors that were tested in 64 common PNs. The first 6 s interval of the trial was divided into 60 bins of 100 ms each, and the average firing rate in each bin (minus the background firing rate over the first 2 s) was calculated over all trials for each cell-odor pair. The binned vectors from all six odors were concatenated for each PN. Principal component analysis was performed on the resulting $360 \times 64$-dimensional matrix using the "pca" routine in MATLAB 2020b. Using the first three principal components, we plotted the values for all time bins separately for each odor, yielding six odor trajectories in a 3D space.

### Odor classification based on PN responses

We performed a classification accuracy analysis[36,106] using the spiking responses of the PN-odor set (64 PNs × 6 odors). As this analysis used the temporal patterns of responses, we ignored the first trial for each cell-odor pair because for some odors the temporal profile of odor delivery was slightly different in the first trial compared to other trials; we used the next six trials for each cell-odor pair. For each trial, the spikes in the response duration (2.1–4.1 s) were divided into eight bins of 250 ms each and the firing rate in each bin was calculated after subtracting the background firing rate. For each odor, we generated a matrix with 6 rows (corresponding to 6 trials) and $64 \times 8$ columns, corresponding to 8 bins from 64 cells. The first 64 columns included the values from the first bins of all 64 cells, the next 64 columns from the second bins, and so on. We estimated classification accuracy for different response durations (i.e., for different numbers of bins used). One trial was selected as test data and the remaining trials were used as training data. To calculate accuracy for a given number of bins (say $k$), we generated a template for each odor by taking the centroid of the training trials using the first $k$ bins. Next, we calculated Euclidean distances between the test data of one odor (using the first $k$ bins) and the templates of all odors. If the test data was closest to the template of the same odor, the accuracy was taken as 1, otherwise 0. The accuracy was calculated for all test odors, and then averaged. This was repeated six times by using a different trial as the test trial each time and the mean accuracy was calculated. We then calculated the cumulative accuracy for different response durations by varying $k$ from 1 to 8.

We also calculated the classification accuracy as a function of the number of cells while using the first three bins (750 ms response duration). We varied the number of cells ($n$) from 5 to 60 in gaps of 5. For each value of $n$, we performed the classification analysis ten times,

each time sampling a random set of $n$ cells, and then took the average accuracy value over the ten iterations.

### Behavior–PN correlations

For each odor, the concentration at which we had the maximum amount of both behavioral and electrophysiological data was used in this analysis. For each pair of odors, we calculated the difference in their preference indices ($\Delta$PI). We generated a response vector for each odor in the pair by concatenating the responses from PNs that were recorded for both odors (again, the response was quantified as the firing rate in the 2.1–4.1 s response duration minus the background firing rate, averaged over all trials). Then, we calculated the Pearson correlation coefficient ($R$) between the two odor vectors and used $1 - R$ as the distance between the PN responses for the odor pair. Next, the PN-behavior correlation was calculated as the Pearson correlation between the $\Delta$PI and $1 - R$ values over all odor pairs. We also calculated the PN-behavior correlation as a function of the number of common PNs ($n$). For a given $n$, we randomly selected $n$ PNs from the common PNs for each odor pair and calculated the PN-behavior correlation from this reduced dataset (this random sampling was repeated 20 times for each $n$ and the average correlation was obtained). For large values of $n$, some odor pairs had fewer than $n$ common PNs and were ignored in the calculations.

### Reporting summary

Further information on research design is available in the Nature Portfolio Reporting Summary linked to this article.

## Data availability

The morphological reconstructions of PNs registered to the template brain are available at https://github.com/neuralsystems/MosquitoAL. Source data are provided with this paper.

## Code availability

The classification analysis was performed using a custom library available at https://github.com/neuralsystems/temporal_classification. The code used for the classification of LNs and PNs based on electrophysiology features and the code used for generating trajectories of PN population responses are available at https://github.com/neuralsystems/NatComm2023[107].

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

## Acknowledgements

We thank Arun Shankar for maintaining the mosquito facility, Anubhav Kumar and Ranjeet Kumar for aiding in behavioral experiments, and Kazumichi Shimizu for advice on electrophysiology and histology. We thank Meg Younger and Leslie Vosshall for sharing the female *Aedes aegypti* brain template on the mosquitobrains.org portal. This work was supported by the DBT/Wellcome Trust India Alliance Fellowship [grant number IA/I/15/2/502091], DST/SERB Swarnajayanti Fellowship [SB/SJF/2021-22/04-C], and SERB Core Research Grant [CRG/2020/004719] awarded to N.G.

## Author contributions

Conceptualization: N.G. and P.S. Methodology: P.S., A.S.B., and N.G. Data collection (electrophysiological): P.S. and S.Go. Data collection (behavioral): P.S., S.Ga., and V.R. Data collection (morphological): P.S., S.Go., and A.K.G. Data curation: P.S. and S.Go. Data analysis: P.S., S.Gu., A.T., A.S.B., S.Ga., and N.G. Writing: P.S., N.G., and A.S.B.

## Competing interests

The authors declare no competing interests.
