## [Peer Review File · Nature Communications]

Combinatorial encoding of odors in the mosquito antennal lobeREVIEWER COMMENTS

Reviewer #1 (Remarks to the Author):

Singh and colleagues have here used how cell patch clamp recordings to investigate the morphological and physiological properties of olfactory projection neurons and local interneurons. The presented work represents what must have been a ginormous work effort, for which the authors should be commended. The morphology and physiology parts are straightforward and executed (as far as I can judge) in accordance with the state-of-the-art. The behavioral experiments, however, are more questionable, and could actually be omitted, without the ms losing all that much.

The ms is well written, easy to follow, and the figures are nice and clean. The analysis methods looks sound to me, except for the part where the behavioral data was incorporated with the physiological data. Glomerular identification is tricky in *Aedes*, and no real consensus has yet emerged. That said, the authors efforts here look appropriate. All in all, an impressive work, that will be read with great interest in the mosquito community!

43: Zika should be capitalized:

124: "From a total of about 1250 mosquito preparations, we were successful in obtaining recordings along with morphological identification from 208 PNs and 53 LNs." Impressive amount of work!

153-155: "The NBLAST scores for homotypic PN pairs were moderately higher than for heterotypic PN pairs (0.57 ± 0.21 vs 0.42 ± 0.24 , $P = 4.9 \times 10^{-19}$, rank sum test; Figure 1e), indicating that homotypic PNs have more similar projections in lateral horn." Im surprised the difference is not greater.

217 ff: Please wrote out the name of the compounds in the main text. PR ACID .01 is not obvious...

249-251: "However, these correlation values had a large spread, indicating that the odor responses of PNs innervating the same glomerulus also varied in many cases." Interesting and a bit surprising no?

334-335: "Thus, even though individual PNs respond promiscuously to odors, large groups of PNs can faithfully encode the odor identity in their temporally patterned responses." The promiscuity of the PNs are likely a result of the high concentration of odors used. Is there are any reason to believe that the PNs in *Aedes* are all that different from those of e.g. *Drosophila*, which have PNS showing a range of tuning width, with partly overlapping, but disntict response spectra?

337 ff: I found the behavioral troublesome. Pretty much all the tested odorants were repellent in the assay, including known attractants. I would hence be very reluctant to draw any conclusions whatsoever from these results. Although the same dilutions were tested in the behavioral and physiological experiments, the actual amount of odor the mosquitoes were exposed to likely differs by several orders

of magnitude. Moreover, I don't fully understand the rationale or the method behind the combination of the behavioral with the physiological data. Or rather, I understand what the authors have done, but I don't see how this would create any meaningful insights.

422-424: "Together, our results show that DEET is detected as an odor in *A. aegypti* and a combinatorial activity in the PN population may be responsible for its aversive nature." Well...I don't think this is the prime reason DEET is so effective. Tone down, or cut out.

Signed
Marcus Stensmyr

Reviewer #2 (Remarks to the Author):

Singh et al "Combinatorial encoding of odors in the mosquito antennal lobe"
NComm Review 220705

This is an interesting and generally well-presented neuronal examination of the antennal lobe (AL) of *Aedes aegypti*. In contrast to the glomerular level focus of most, not all, other AL studies in mosquitoes, the most interesting component of Singh et.al. is the use of patch-clamp based approaches to focus on individual projection neurons (PN) and local neurons (sometimes known as lateral neurons, LNs). This study is a welcome departure from that glomerular-centric paradigm.

The authors establish a viable setup for whole-cell AL patch-clamping coupled with dye-filling morphological analyses will be of interest to others in the field. They use this approach to sample a meaningful population of AL neurons with the caveat that limitations in the preparation (only able to access part of the AL) and data analyses (see below) have likely biased their interpretations, perhaps significantly. They also use the dye-filling component of their experimental paradigm to partially map AL neuronal connections as well as a collection of axonal projections to higher brain centers such as the mushroom body. However, because this type of morphological mapping has also been done in *Drosophila* (where genetic tools and other elements provide better resolution) compared to the ePhys data set this component is largely underwhelming.

While there is a lot to like here especially the novel ePhys paradigm my enthusiasm for the study is largely curtailed by several concerns (see below), most notably the reliance on a limited number of odorants at only one concentration. In any case that the authors arrive at a combinatorial coding model for AL neurons is hardly surprising and certainly not unique.

Strengths

- Patch clamping of AL neurons allows for more detailed assessment of odor evoked activity/coding at the discrete levels of PN and LNs.

- Population level analyses of these recordings yield meaningful data sets that are used to make interesting assessments of odor representations/coding that is mapped to behavioral valence.
- Even though the AL3/PL2 PN studies of palpal responses may be a bit speculative regarding the inter-neuronal excitation via a LN they are still interesting and suggest these can be linked directly to the well-characterized palpal Or8/49 expressing ORNs-it would be interesting to see if these could be evoked by the odorants known to activate those palpal ORNs.

Concerns/Weaknesses:

- My most significant concern that impacts my overall review is that the odor stimulations for ePhys and behavior was carried out for only 14 odorants and most importantly at only a single concentration. While I can live with the very modest spectrum of odorants that are used, the lack of even a modest (3-4 concentrations) dose response component for the ePhys dramatically undercuts the value of that odorant-limited data set and all of the downstream analyses from which the combinatorial coding thesis at the heart of this study is generated from.
- Less than 25% of preparations yielded viable recordings/morphology data, potentially resulting in skewed data. For example, in that light I am not sure I am willing to embrace the report of rare (7/208) instances of multi-glomerular PNs (line 133). While not ruling out the potential for this I would need a significantly larger data set.
- I am also concerned about the need to inject “a small constant current” (line 472) to bring resting potential between -45 and -60mV. What is the normal resting potential?
- The value/utility of using ePhys to classify AL neurons (figure 3) is not clear. This limitation is impacted by not fully appreciating how those responses are compared across odorants where response profiles (and concentrations) may be significantly different.
- Because of limited sampling and other considerations, I not convinced about the value/interpretation of sub-threshold LN responses (Fig 5d) especially in light of lack of dose response.
- The all or none response aspects (aversive or attractive) of the behavioral assay is troubling. If a mosquito is just beyond the sliding door choice point it is counted the same as an individual which is robustly attracted or repelled such that it is fully shifted towards (or away) from the odor source.
- I am also troubled by lines 544-546, describing the behavioral methods. “experiments in which experimental conditions or the number of responding mosquitoes were out of the expected range were excluded”. What does this mean? Was there an expected range???
- The rationale presented in the discussion seems out of order to me. I would start at line 383 and include behavior correlation after ePhys.

Reviewer #3 (Remarks to the Author):

This paper is an electrophysiological tour de force that expands our understanding of the organization and function of the *Aedes aegypti* antennal lobe. It is complementary to other studies that have relied on imaging. The exploration of the PNs and LNs in the antennal lobe of the mosquito is an area where

little is known. This study expands our understanding of how odor information may be processed by the mosquito. I am particularly impressed by the connectivity suggested by the morphological and electrophysiological data presented. This is an important paper that will help inform our understanding of the mosquito olfactory processing both now and in the future. This paper should be of interest to a wide audience.

The characterization of the PNs and LNs is quite beautiful and clear. It is exciting that the authors can provide evidence that odors are easier for mosquitoes to discriminate as the number of PNs that are involved increases. The paper shows clear support for the combinatorial coding of odors in the mosquito. This work opens the door for studies that show the direct connections between ORNs, PNs, and LNs that will likely require genetically encoded methods to achieve.

The methods and data presented in the paper are clear. I provide a few important comments for the authors to consider to improve the manuscript.

Major Comments

In the abstract, I do not think this is the first detailed description of olfactory neurons in the mosquito CNS. Olfactory receptor neuron responses in the antennal lobe have been well described previously. However, it is a detailed description that covers new ground. I would remove the word first from that sentence in the abstract (line 37).

In Figure 1, the authors present representative images of the PN types found. It is possible to present more or all of the PN morphological images in a supplementary figure? As our understanding of the glomeruli increases, this data may be useful to connect these to a newer reconstruction of the mosquito brain if necessary.

The discussion of the DEET results was incomplete and the data does not support their interpretation. The authors do show that DEET is mildly repellent in their behavioral assay, however this does not reflect the strength of the repellency that has been shown in the context of human host odor which contains multiple volatiles. The conclusions drawn by the authors do not take into account other possibilities that may exist. This is in part due to the fact that the authors did not test DEET in the context of other odors. A controversy in the field still exists about whether DEET acts alone or requires additional odors to achieve full effect. Thinking about the data in light of the confusant hypothesis would be helpful. I suggest a few reviews and papers that will help the authors take another look at their interpretation of the DEET results.

Dogan, E.B., Ayres, J.W., Rossignol, P.A., 1999. Behavioural mode of action of DEET: inhibition of lactic acid attraction. *Med. Vet. Entomol.* 13, 97–100.

Pellegrino, M., Steinbach, N., Stensmyr, M.C., Hansson, B.S., Vosshall, L.B., 2011. A natural

polymorphism alters odour and DEET sensitivity in an insect odorant receptor. *Nature* 478, 511–514.

DeGennaro, M., McBride, C.S., Seeholzer, L., Nakagawa, T., Dennis, E.J., Goldman, C., Jasinskiene, N., James, A.A., Vosshall, L.B., 2013. Orco mutant mosquitoes lose strong preference for humans and are not repelled by volatile DEET. *Nature* 498, 487–491.

DeGennaro, M., 2015. The mysterious multi-modal repellency of DEET. *Fly (Austin)* 9, 45–51.

CHAPTER 7 - Multimodal mechanisms of repellency in arthropods : <https://doi.org/10.1016/B978-0-323-85411-5.00005-4>

I understand that this is not a focus of the paper, but I strongly suggest the authors take a more nuanced look at how DEET could be acting in light of what has been published and their own data set.

Minor Comments

Figure 3i is a useful panel, however, its utility is poorly described. Can you be more clear about the two clusters in the text? It is interesting that two PNs cluster with the LN predominant cluster. Also a few LNs cluster with PN dominant cluster. Could you explain this more in the paper? I would like a little more context for this figure panel in the results section (just a sentence or two). It also may be worth discussing these results in the discussion.

Labeling 3i with PN-enriched and LN-enriched for the respective clusters would improve the figure for the reader.

Figure 8a is a bit confusing. The diagram should clearly show how the sliding door separates the assay for scoring. The dotted line points to a box, not a sliding door. I am assuming the entire box moves into the main chamber? What is the black line signifying? Please discuss the black line in the legend if it is necessary for understanding the assay. This is an interesting assay and a little improvement to the diagram is worthwhile.

Reviewer comments are in black. Our replies are in blue.

Reviewer #1 (Remarks to the Author):

Singh and colleagues have here used how cell patch clamp recordings to investigate the morphological and physiological properties of olfactory projection neurons and local interneurons. The presented work represents what must have been a ginormous work effort, for which the authors should be commended. The morphology and physiology parts are straightforward and executed (as far as I can judge) in accordance with the state-of-the-art. The behavioral experiments, however, are more questionable, and could actually be omitted, without the ms losing all that much.

The ms is well written, easy to follow, and the figures are nice and clean. The analysis methods looks sound to me, except for the part where the behavioral data was incorporated with the physiological data. Glomerular identification is tricky in *Aedes*, and no real consensus has yet emerged. That said, the authors efforts here look appropriate. All in all, an impressive work, that will be read with great interest in the mosquito community!

We thank the reviewer for highlighting the strengths of our work and providing many insightful comments below, which have been addressed in this revision.

43: Zika should be capitalized

We have corrected this.

124: "From a total of about 1250 mosquito preparations, we were successful in obtaining recordings along with morphological identification from 208 PNs and 53 LNs." Impressive amount of work!

We thank the reviewer for acknowledging the efforts of the team members who generated the data over 6 years.

153-155: "The NBLAST scores for homotypic PN pairs were moderately higher than for heterotypic PN pairs (0.57 ± 0.21 vs 0.42 ± 0.24 , $P = 4.9 \times 10^{-19}$, rank sum test; Figure 1e), indicating that homotypic PNs have more similar projections in lateral horn." I'm surprised the difference is not greater.

In terms of absolute values, the difference appears to be small, and this can partially be attributed to the fact that the homotypic PNs were recorded in different samples and so the NBLAST scores can be lower than expected due to some noise in image registrations, and variable sample integrity. However, despite these factors, the difference is statistically very reliable and we believe the effect size. ~ 0.15 , is meaningful. In general, PNs as a class are very similar; within that class, homotypic PNs are even more similar.

We have now extended the anatomical work in this study by analyzing the lateral horn (LH) projections of different uniglomerular PNs. We found that the PNs innervating dorsomedial glomeruli within the antennal lobe mostly project to dorsal region within LH, while the other

PNs mostly project to ventral-anterior region in the LH. This result has now been included in Figure 1c. Further, we compared the odor responses of the two groups of PNs and found that PN pairs innervating the dorsomedial glomeruli show higher positive correlations in their responses than PN pairs innervating other glomeruli. This result has now been included as Supplementary Figure 4c. We have also modified Supplementary Figure 4a to show homotypic PN-groups innervating dorsomedial region in red, and other regions in blue.

217 ff: Please write out the name of the compounds in the main text. PR ACID .01 is not obvious...

We have implemented this suggestion and replaced the non-standard odor acronyms with full names at all places in the text.

249-251: "However, these correlation values had a large spread, indicating that the odor responses of PNs innervating the same glomerulus also varied in many cases." Interesting and a bit surprising no?

Indeed, we were also surprised by the observed spread in the correlations between homotypic PN pairs, which is the reason why we wrote the sentence that the reviewer has quoted. Some homotypic PN groups (such as I-AL3, I-AM1, I-AM4, I-AC2) show a high positive correlation between odor responses while others do not (**Supplementary Figure 4a**). We have mentioned two possible reasons for this spread in the text: differences between differential inputs received by different PNs innervating the same glomerulus (indicating functional diversity among homotypic PNs in mosquitoes) or the differences between the individuals from which the PN recordings were obtained.

Another possible reason could have been errors in the identification of glomeruli, but we can rule out this as an explanation of the observed variability in odor responses to a large extent. Two members of our team had independently labeled each histological image. They independently assigned a confidence score (0-5) to each PN based on comparing the location, the shape, the size, and the neighborhood of the labeled glomerulus with the atlas (Ignell et al. 2005). Supplementary Figure 4b includes data from 64 uniglomerular PNs (including 74 homotypic PN pairs) that were morphologically identified. To test if incorrect glomeruli identification is the reason for the observed variability among homotypic PNs, we repeated the analysis by putting a cut-off of 3.5 on the average confidence score from the two experimenters: this resulted in 51 PNs and 39 homotypic pairs (about half of the original set of homotypic pairs). The plot below shows the correlations between odor responses among these homotypic PN pairs. We still see the same level of variability as in the original figure. Thus, removing half of the data with relatively less confident glomerular identifications did not reduce the variability among homotypic PNs.

334-335: “Thus, even though individual PNs respond promiscuously to odors, large groups of PNs can faithfully encode the odor identity in their temporally patterned responses.” The promiscuity of the PNs are likely a result of the high concentration of odors used. Is there are any reason to believe that the PNs in *Aedes* are all that different from those of e.g. *Drosophila*, which have PNs showing a range of tuning width, with partly overlapping, but distinct response spectra?

We did not mean to claim here that mosquito PNs are different from fly PNs in tuning widths. We agree with the reviewer that PNs exhibit a range of tuning widths. We have now modified the corresponding phrase to “*individual PNs respond with different levels of promiscuity to odors*”. Please note that the odor classification accuracy increased with the size of the PN population (Figure 7), and thus we are justified in claiming that the large groups of PNs are required to faithfully encode odor identity.

337 ff: I found the behavioral troublesome. Pretty much all the tested odorants were repellent in the assay, including known attractants. I would hence be very reluctant to draw any conclusions whatsoever from these results. Although the same dilutions were tested in the behavioral and physiological experiments, the actual amount of odor the mosquitoes were exposed to likely differs by several orders of magnitude. Moreover, I don't fully understand the rationale or the method behind the combination of the behavioral with the physiological data. Or rather, I understand what the authors have done, but I don't see how this would create any meaningful insights.

Of the 14 monomolecular odors tested in our assay, 1 was attractive, 7 were repulsive and 6 were neutral. We agree that some of the 7 odorants that we found to be repulsive have been reported to be attractive in previous studies. But this difference is likely because of the differences in odor concentrations used in different studies, and not because of a bias in our assay towards negative preferences (we had taken care to keep the two arms identical in terms of appearance and ambient cues, and also switched the odor and the control sides in different trials to further eliminate any possibility of bias). To confirm this, we have now performed behavioral experiments with lower concentrations of 1-octen-3-ol (1OCT3OL) and 6-methyl-5-hepten-2-one (6MHO). Although these odorants were aversive at 0.01 concentrations reported in the manuscript, at the lower concentrations they showed positive preference indices in the same assay, as shown below. Thus, we are confident that our

assay is not biased to result in negative preferences. We have included these results in new Supplementary Figure 9.

As we described in our response to Reviewer 2, we have only limited electrophysiological data for lower concentrations of odors (as our priority was to obtain data for multiple odors and with at least 6 trials per odor). These data are now added to Figures 4b, 4d, 5b, 5c, 7a, and 7b in the manuscript.

The second concern expressed by the reviewer is whether the amount of odor experienced by the mosquitoes differed by several orders of magnitude between the behavioral and the electrophysiological experiments. To check this, we measured the odor concentration in both experimental setups by using a photoionization detector (PID) in place of a mosquito. The image below shows the PID signals recorded in the two set-ups. The first panel shows PID signal for 6MHO .01 in the electrophysiology rig: the PID probe was kept at the same position where the mosquito would normally be kept. Grey traces (almost overlapping) are different trials; the average of all the trials is shown in black. The second panel shows the PID signal recorded for the same odor (6MHO .01) when a small tube connected to the PID was placed inside the behavioral chamber in the odor arm. In behavioral experiments, odor was delivered in pulses of 1s ON:4s OFF for 5 minutes. The amplitude of the PID signal is comparable (around 60 mV) in both graphs, suggesting that the finally delivered concentrations in the two assays were similar. We have included this plot as Supplementary Figure 8.

Our rationale for combining behavioral and electrophysiological data was to check the extent to which the behavioral preference is detectable in the PN activity. While we did not observe a very simpler marker for attractive or aversive preference (in terms of overall levels of activity or an obvious spatial segregation among glomeruli), our results in Figure 8 show that odors with similar activity in the PN population tend to generate similar behavioral preference, and that this relationship is stronger when more PNs are considered in the population. We believe these results are informative and support the idea of population coding of odors in the mosquito antennal lobe.

422-424: “Together, our results show that DEET is detected as an odor in *A. aegypti* and a combinatorial activity in the PN population may be responsible for its aversive nature.”
Well...I don't think this is the prime reason DEET is so effective. Tone down, or cut out.

We have rephrased this sentence in the Discussion section to a more modest and factual version (without making a claim about the reason for its effectiveness): “*Together, our results show that DEET is detected as an odor in A. aegypti and activates multiple PNs in the antennal lobe.*”

Reviewer #2 (Remarks to the Author):

This is an interesting and generally well-presented neuronal examination of the antennal lobe (AL) of *Aedes aegypti*. In contrast to the glomerular level focus of most, not all, other AL studies in mosquitoes, the most interesting component of Singh et.al. is the use of patch-clamp based approaches to focus on individual projection neurons (PN) and local neurons (sometime known as lateral neurons, LNs). This study is a welcome departure from that glomerular-centric paradigm. The authors establish a viable setup for whole-cell AL patch-clamping coupled with dye-filling morphological analyses will be of interest to others in the field. They use this approach to sample a meaningful population of AL neurons with the caveat that limitations in the preparation (only able to access part of the AL) and data analyses (see below) have likely biased their interpretations, perhaps significantly. They also

use the dye-filling component of their experimental paradigm to partially map AL neuronal connections as well as a collection of axonal projections to higher brain centers such as the mushroom body. However, because this type of morphological mapping has also been done in *Drosophila* (where genetic tools and other elements provide better resolution) compared to the ePhys data set this component is largely underwhelming. While there is a lot to like here especially the novel ePhys paradigm my enthusiasm for the study is largely curtailed by several concerns (see below), most notably the reliance on a limited number of odorants at only one concentration. In any case that the authors arrive at a combinatorial coding model for AL neurons is hardly surprising and certainly not unique.

We thank the reviewer for evaluating our manuscript and providing helpful suggestions below.

Strengths

- Patch clamping of AL neurons allows for more detailed assessment of odor evoked activity/coding at the discrete levels of PN and LNs.
- Population level analyses of these recordings yield meaningful data sets that are used to make interesting assessments of odor representations/coding that is mapped to behavioral valence.
- Even though the AL3/PL2 PN studies of palpal responses may be a bit speculative regarding the inter-neuronal excitation via a LN they are still interesting and suggest these can be linked directly to the well-characterized palpal Or8/49 expressing ORNs-it would be interesting to see if these could be evoked by the odorants known to activate those palpal ORNs.

We thank the reviewer for highlighting the strengths of our study. We also thank the reviewer for bringing up the question of whether AL3/PL2 PNs are activated by the odors that activate palp ORNs. We checked this, and found that AL3/PL2 PNs did not respond to the odors that are known to activate maxillary palps, suggesting that they don't receive direct input from the palp sensory neurons. This is consistent with previous reports that only MD1, MD2, and MD3 glomeruli receive direct inputs from the palp. We have now added the above results as a new Supplementary Figure 6b, and added the following description to the text: "*These glomeruli do not receive direct sensory input from the palps (Ignell et al., 2005; Herre et al., 2022)), and expectedly did not respond to odors that are known (Singh et al, biorxiv) to activate the sensory neurons in the palp (Supplementary Figure 6b). But they still showed spontaneous spikes, suggesting indirect inputs from within the antennal lobe (Figure 6c)*".

My most significant concern that impacts my overall review is that the odor stimulations for ePhys and behavior was carried out for only 14 odorants and most importantly at only a single concentration. While I can live with the very modest spectrum of odorants that are used, the lack of even a modest (3-4 concentrations) dose response component for the ePhys dramatically undercuts the value of that odorant-limited data set and all of the downstream analyses from which the combinatorial coding thesis at the heart of this study is generated from.

We agree that a larger odor panel, with more concentrations of each odorant, is always desirable. The only reason we had to limit the size of the odor panel was that the intracellular recordings in the mosquito proved to be very challenging and we had limited time available

with each cell for testing odors. Further, while some studies in the field of olfaction test each odor in 1-3 trials, we tried to obtain at least 6 trials for every odor, to increase the reliability of the observed odor responses, but this further reduced the number of odorants that we could test. Despite these limitations, we did manage to obtain data at more than one concentration for some odorants. For the quantitative analyses in the manuscript, we included data for only one concentration at which we had the maximum data (typically the highest concentration). However, based on the reviewer's feedback, we have now shown data for lower concentrations of some odorants (tested in at least 10 cells each) in Figures 4b, 4d, 5b, 5c, 7a and 7b. We also summarize one take-away from these data in the figure below: the fractions of PNs activated by the odorants at 10-fold lower concentrations (blue) are comparable to the fractions at the higher concentrations (black). Thus, these additional results at lower concentrations of odorants further support the idea of a combinatorial code.

Fractions of PNs activated by different odorants. The number above each bar shows the number of PNs in which the corresponding odorant was tested (this figure is an excerpt from the revised Figure 7a in the manuscript).

Less than 25% of preparations yielded viable recordings/morphology data, potentially resulting in skewed data. For example, in that light I am not sure I am willing to embrace the report of rare (7/208) instances of multi-glomerular PNs (line 133). While not ruling out the potential for this I would need a significantly larger data set.

Although the success rate of the experiments was low (because of the challenging nature of the experiments), we compensated for it by increasing the number of experiments, so that in the end we had a sufficiently large number of uniglomerular PNs (more than 200) for our quantitative analysis. We agree with the reviewer that our sample size for multiglomerular PNs is small (7), and therefore we have not made any claims about their responses in the paper.

After reading the reviewer's comment, we realize that our writing might have given the impression that mosquitoes have a uniquely small fraction of multiglomerular PNs compared to *Drosophila*, which should not be concluded from this data (not so much because of the low success rate in the experiments as it was compensated by a large number of experiments, but because of the fact that we could target cell bodies in only anterodorsal and dorsolateral clusters around the AL but not in the ventral cluster, which may have a

higher proportion of multiglomerular PNs). We have now added clarifications for this point, both in the Results and in the Discussion.

In the Results, we now write, *“By examining the dendritic innervations within the AL, we found that 201 of 208 recorded PNs were uniglomerular, with dense and complete innervation of the corresponding glomerulus (Figure 1b and Supplementary Figure 1a); this high proportion of uniglomerular PNs may be because of the targeting of mostly dorsal clusters of cell bodies.”*

In the Discussion, we now write, *“however, the low number of multiglomerular PNs in our dataset this may be related to the fact that cell bodies in ventral AL were not accessible in our preparation. Further work targeting cell bodies in all clusters around the AL is needed to know the exact proportion of multiglomerular PNs in mosquitoes.”*

I am also concerned about the need to inject “a small constant current” (line 472) to bring resting potential between -45 and -60mV. What is the normal resting potential?

The resting membrane potential for recordings with high-resistance seals in our experiments was typically observed to be between -50 and -60 mV. However, in some cases the resting membrane potential was somewhat lower or higher. It is a common occurrence in whole-cell patch-clamp recordings, and in such cases it is an accepted practice to inject a small amount of hyperpolarizing or depolarizing current to bring the membrane potential in the normal range and stabilize the recording, and has been reported in many studies: for example, Chou et al., 2010, *Nature Neuroscience*; Seki et al., 2017, *BMC Biology*; Shimizu & Stopfer, 2017, *Frontiers in Neural Circuits*; Wilson et al., 2004, *Science*; Zhang & Gaudry, 2016, *ELife*. We restricted the amount of the injected current to less than 0.03 nA whenever it was used (we have now added this information in the Methods). Please note that a constant current injection affects only the baseline activity of a neuron to a small degree and does not change the fact whether the activity increases or decreases at the time of odor stimulus (we compute an odor response as the difference between the activities in the odor period and the background period).

The value/utility of using ePhys to classify AL neurons (figure 3) is not clear. This limitation is impacted by not fully appreciating how those responses are compared across odorants where response profiles (and concentrations) may be significantly different.

The rationale for using electrophysiological parameters to classify AL neurons is that: (1) it confirms that PNs and LNs have systematic differences in the membrane properties; and (2) it provides a simple way for future studies to identify whether a recorded neuron is a PN or an LN, even if the morphological information is not available for that neuron (in cases where dye-fills are not available or successful, or in cases where multiple neurons are recorded in the same antennal lobe and it is not possible to identify them separately using just one or two fluorescent dyes that may be available for use). We have now added a new sentence to clarify the utility of this analysis in the Results: *“The high classification accuracy confirms that PNs and LNs have systematic differences in their electrophysiological properties, and that it is possible to use these differences to identify a recorded AL neuron as an LN or a PN even without knowing the morphology of the neuron.”*

In the classification analysis, we wanted to use those electrophysiological features that are properties of the recorded neuron and not of the stimulus. These features would allow one to identify neurons as LNs or PNs independently of the stimulus, and then one can compare the odor responses of these two types of neurons. However, if one already uses the information about odor responses in differentiating LNs and PNs, then one cannot use the same neurons to comment on the odor responses of the two classes of neurons (as it would become a circular analysis). Keeping this in mind, we used only those electrophysiological features that depend on the fundamental membrane properties of a neuron and should be independent of the stimulus identity or concentration (e.g., the spike amplitude). In our analysis presented in the manuscript, we have extracted these basic electrophysiological features for each neuron using all the recordings available for that neuron. To confirm that the extracted features and the classification are not dependent on odor responses, we performed this analysis again by excluding spikes in the odor response duration from each trial. The plots below show the results for this revised analysis: the values of ephys features in figures **a-d** below are very similar to **Figures 3c-f** in the manuscript, and the classification in figure **e** below is very similar to **Figure 3i** in the manuscript. The accuracy of the classification remains unchanged. Thus, we are confident that our classification based on ephys features is independent of the stimulus, as we intended it to be.

Because of limited sampling and other considerations, I not convinced about the value/interpretation of sub-threshold LN responses (Fig 5d) especially in light of lack of dose response.

The sub-threshold LN responses in Fig. 5d were included to illustrate that an LN may receive weak input during an odor stimulus, which affects the neuron reliably (as it produces an EPSP at the same time relative to the stimulus in different trials) even though it does not always result in spikes. To further support this point, we have now expanded Fig. 5d by including recordings from one more LN: this cell also responds with spikes during odor onset in some trials whereas in other trials it just shows excitatory sub-threshold activity for the same odor.

The all or none response aspects (aversive or attractive) of the behavioral assay is troubling. If a mosquito is just beyond the sliding door choice point it is counted the same as an individual which is robustly attracted or repelled such that it is fully shifted towards (or away) from the odor source.

Most behavioral assays for assessing odor preference ultimately rely on a binary (i.e., all-or-none) classification of the animal's choice, depending on whether the animal moves beyond a certain threshold point in the arena. Even if we set the threshold points close to the ends of two arms, as suggested by the reviewer, it would still remain an all-or-none assay. In our view, the real concern is to find out what is the ideal threshold point for the assay. While we agree that are some advantages in keeping the threshold points closer to the ends, there are also some disadvantages in doing so, as we describe below.

1. There is a trade-off between the reliability of individual responses and the number of responding mosquitoes. We agree that keeping the threshold points towards the ends of the arms would make the choice of an individual mosquito more reliable (as indicated by the reviewer). But, in our kind of assay, where groups of mosquitoes are tested together, the number of mosquitoes that are counted as responding would reduce in this case (as there will be no difference between a mosquito that remains in the release chamber and a mosquito that moves half-way into one of the arms – both would be counted as non-responding). And such reduction in the number of responding mosquitoes would, in turn, reduce the statistical reliability of the group assay. Thus, there is also a disadvantage in the keeping the threshold points close to the ends.

2. For an attractive odor, a mosquito would be motivated to reach the odor source and is therefore expected to move till the end of the odorized arm. For an aversive odor, a mosquito would be motivated to avoid the odor, which can be accomplished by moving just a little bit into the control arm; in this case, the mosquito will have very little incentive to move further deep into the control arm. Thus, keeping the threshold points closer to the ends would make it difficult to detect aversive behavioral preferences. As we have mentioned in the Discussion, our design of the behavioral assay was guided by the consideration that the assay should be able to detect both attractive and aversive preferences with equal ease (we think that some of the previously used assays in the field are biased against aversive preferences).

We further note that the sliding door compartment was 12 cm wide, and it included two connected doors, one per each arm (6 cm inside the arms, which is not very small compared to the body-length of a mosquito). This aspect of the design was not very clear in our original schematic and has now been improved in **Figure. 8a** and also explicitly mentioned in the revised Discussion. Thus, a mosquito was counted as responding only if it finally moved at

least 6 cm into an arm and remained there at the end of the 300-second experiment. A mosquito that went into an arm but came out and was found in the central area at the end of the experiment would not be counted.

Considering all these factors, we believe that our assay design provides a good trade-off between the reliability of individual choices and the group-choice and the ability to measure both attractive and aversive preferences. We also note that similar assays have been used previously in flies (e.g., Suh et al., 2004, *Nature*), moths (e.g., Najar-Rodriguez et al, 2010, *Journal of Experimental Biology*) and mosquitoes (e.g., Vinauger et al., 2018, *PNAS*), where the threshold point has been set close to the intersection of the two arms rather than at the ends of the arms.

I am also troubled by lines 544-546, describing the behavioral methods. “experiments in which experimental conditions or the number of responding mosquitoes were out of the expected range were excluded”. What does this mean? Was there an expected range???

We realize that our description of this part was fragmented and thank the reviewer for pointing this out. At an earlier place in the Methods section, we had reported that the temperature inside the chamber was maintained between 25°C-30°C and the relative humidity was between 45%-80%. However, in very few experiments, the temperature or humidity were out of the above-mentioned ranges. Those experiments were excluded from the analysis. Apart from this, the experiments in which less than 5 mosquitoes made a choice were also excluded.

We have now rewritten this part in the Methods to make it more cohesive and clear: “*The temperature inside the chamber was maintained between 25-30°C and the relative humidity between 45-80%. Any experiment in which the temperature or the humidity inside the chamber was out of this range or the number of responding mosquitoes was less than 5 was excluded from the analysis.*”

The rational presented in the discussion seems out of order to me. I would start at line 383 and include behavior correlation after ePhys.

We have now improved the flow of the Discussion by rearranging the text as suggested by the reviewer.

Reviewer #3 (Remarks to the Author):

This paper is an electrophysiological tour de force that expands our understanding of the organization and function of the *Aedes aegypti* antennal lobe. It is complementary to other studies that have relied on imaging. The exploration of the PNs and LNs in the antennal lobe of the mosquito is an area where little is known. This study expands our understanding of how odor information may be processed by the mosquito. I am particularly impressed by the connectivity suggested by the morphological and electrophysiological data presented. This is an important paper that will help inform our understanding of the mosquito olfactory processing both now and in the future. This paper should be of interest to a wide audience.

The characterization of the PNs and LNs is quite beautiful and clear. It is exciting that the authors can provide evidence that odors are easier for mosquitoes to discriminate as the number of PNs that are involved increases. The paper shows clear support for the combinatorial coding of odors in the mosquito. This work opens the door for studies that show the direct connections between ORNs, PNs, and LNs that will likely require genetically encoded methods to achieve.

The methods and data presented in the paper are clear.

We thank the reviewer for appreciating the merits of our work and suggesting improvements to the manuscript.

In the abstract, I do not think this is the first detailed description of olfactory neurons in the mosquito CNS. Olfactory receptor neuron responses in the antennal lobe have been well described previously. However, it is a detailed description that covers new ground. I would remove the word first from that sentence in the abstract (line 37).

We have implemented this suggestion. We now write, '*Our results provide a detailed description of the second-order olfactory neurons in the central nervous system of mosquitoes...*'.

Although we can justify the use of the word "first" to describe our work on "second-order olfactory neurons", we have dropped it now as a matter of style.

In Figure 1, the authors present representative images of the PN types found. It is possible to present more or all of the PN morphological images in a supplementary figure? As our understanding of the glomeruli increases, this data may be useful to connect these to a newer reconstruction of the mosquito brain if necessary.

We have now added a new Supplementary Figure 1a showing morphologies of additional uniglomerular PNs.

The discussion of the DEET results was incomplete and the data does not support their interpretation. The authors do show that DEET is mildly repellent in their behavioral assay, however, this does not reflect the strength of the repellency that has been shown in the context of human host odor which contains multiple volatiles. The conclusions drawn by the authors do not take into account other possibilities that may exist. This is in part due to the fact that the authors did not test DEET in the context of other odors. A controversy in the field still exists about whether DEET acts alone or requires additional odors to achieve full effect. Thinking about the data in light of the confusant hypothesis would be helpful. I suggest a few reviews and papers that will help the authors take another look at their interpretation of the DEET results.

Dogan, E.B., Ayres, J.W., Rossignol, P.A., 1999. Behavioural mode of action of DEET: inhibition of lactic acid attraction. *Med. Vet. Entomol.* 13, 97–100.

Pellegrino, M., Steinbach, N., Stensmyr, M.C., Hansson, B.S., Vosshall, L.B., 2011. A natural polymorphism alters odour and DEET sensitivity in an insect odorant receptor. *Nature* 478, 511–514.

DeGennaro, M., McBride, C.S., Seeholzer, L., Nakagawa, T., Dennis, E.J., Goldman, C., Jasinskiene, N., James, A.A., Vosshall, L.B., 2013. Orco mutant mosquitoes lose strong preference for humans and are not repelled by volatile DEET. *Nature* 498, 487–491.

DeGennaro, M., 2015. The mysterious multi-modal repellency of DEET. *Fly (Austin)* 9, 45–51.

CHAPTER 7 - Multimodal mechanisms of repellency in arthropods
: <https://doi.org/10.1016/B978-0-323-85411-5.00005-4>

I understand that this is not a focus of the paper, but I strongly suggest the authors take a more nuanced look at how DEET could be acting in light of what has been published and their own data set.

We thank the reviewer for suggesting these references, which are now included in our Discussion. We have also added: “*There may also be additional mechanisms for the action of DEET, such as interactions with host odors*”. Further, as we noted in our response to Reviewer 1, we limit our claim to the fact that DEET activates multiple PNs in the antennal lobe. We believe this is still useful information and will inform future studies that focus on understanding the neural mechanisms of DEET’s repellence.

Figure 3i is a useful panel, however, its utility is poorly described. Can you be more clear about the two clusters in the text? It is interesting that two PNs cluster with the LN predominant cluster. Also a few LNs cluster with PN dominant cluster. Could you explain this more in the paper? I would like a little more context for this figure panel in the results section (just a sentence or two). It also may be worth discussing these results in the discussion. Labeling 3i with PN-enriched and LN-enriched for the respective clusters would improve the figure for the reader.

We thank the reviewer for the suggestion. We have labeled the clusters as ‘PN-enriched cluster’ and ‘LN-enriched cluster’. In the text, we have summarized the differences between PNs and LNs: “*on average, PNs had fewer isolated spikes and their spikes were smaller, wider, and had less after-hyperpolarization compared to LN spikes*”.

We have also clarified the text and mentioned the reasons for the inaccurate classifications observed in 8 out of 212 cases. We now write, “*An unsupervised hierarchical clustering performed using the four properties was able to group the neurons into two broad clusters, which we labeled as ‘PN-enriched’ and ‘LN-enriched’ clusters as they matched the morphological identification of PNs and LNs with 95% accuracy (Figure 3i). We observed that out of 212 neurons analyzed, only 2 PNs and 6 LNs were placed in the wrong clusters: the 2 PNs had unusually large fractions of isolated spikes (0.80 and 0.56 respectively, compared to the group mean of 0.04), while the LNs that were misclassified had unusually large spike half-widths (2 cases), small after-hyperpolarization (1 case), or small spike amplitudes (3 cases).*”

Figure 8a is a bit confusing. The diagram should clearly show how the sliding door separates the assay for scoring. The dotted line points to a box, not a sliding door. I am assuming the entire box moves into the main chamber? What is the black line signifying? Please discuss

the black line in the legend if it is necessary for understanding the assay. This is an interesting assay and a little improvement to the diagram is worthwhile.

We have now improved the illustration in Figure 8a and highlighted the sliding door using a different color (brown). We have also added a top view of the chamber for more clarity. The sliding door is shown half-way inside the main chamber to clarify how it compartmentalizes the main chamber when it is pushed inside. The black line is the handle for the rotating door (now clearly marked in the revised figure).

REVIEWER COMMENTS

Reviewer #1 (Remarks to the Author):

The authors have done a good job addressing the concerns I had. Again, overall I think this a very well executed story, which Im sure will be recieved with much interest in the mosquito community. The sentences describing the action of DEET though should be revised. "There may also be additional mechanisms for the action of DEET...", should read "There are, however, additional mecahanisms through which DEET function in Aedes", or something along those lines.

Reviewer #2 (Remarks to the Author):

Singh etal "Combinatorial encoding of odors in the mosquito antennal lobe"
NComm Review of Revised Manuscript

As was the case for the initial submission this report is an interesting and generally well-presented neuronal examination of the antennal lobe (AL) of *Aedes aegypti* that has now been significantly improved upon revision.

As before, there is much to like here and the authors have generally (apart from ongoing concerns, detailed below) responded to previous issues. Rather than detail (once again) those positives I shall instead focus on areas that remain significantly concerning.

1. As in my initial review, I remain deeply concerned with the near complete use of a single (relatively, high) concentration for odorants used throughout this study. The revision now provides what can only be described as a token gesture of 1 additional concentration for 'some' odorants (3) and 2 additional concentrations for a single odorant (1oct3ol). This concern is exacerbated as it appears those additional odor concentrations were not actually utilized very much at all as the vast majority of PN/LN recordings displayed in Figures 4 and 5, display a grey value which, according to the legend "indicates that the odor was not tested..."

2. I am also concerned that the ePhys studies seem (perhaps I am not reading it correctly) to have not been normalized/corrected for solvent (mineral oil/water) related responses. From lines 701-704, it appears that only background using air from a 'clean empty' vial (Line 614) was used for data analyses.

3. Despite the additional information in this revised submission, I remain skeptical about the value of the behavioral setup used here. The airflow (5L/min from each side!) was significantly higher than in ePhys (2L/min). It's strange from the PID pulses provided in supplemental data there doesn't appear to be any mixing-was this tube placed just at the odor delivery point??

4. Even with those flaws it is hardly surprising to find a correlation between behavioral and ePhys data for PNs, I was more interested to see if there was a correlation (or not) to LN activity which was neither presented nor discussed??

Overall, I remain underwhelmed by these concerns regarding the experimental rigor of this otherwise

intriguing report.

Reviewer #3 (Remarks to the Author):

I have even more enthusiasm for the manuscript after the the authors' revision. I appreciate that the concerns I had about some aspects of the manuscript have been appropriately addressed. The manuscript is now ready for publication and will make a significant contribution to our understanding of mosquito olfaction.

Reviewer #1 (Remarks to the Author):

The authors have done a good job addressing the concerns I had. Again, overall I think this a very well executed story, which I'm sure will be received with much interest in the mosquito community. The sentences describing the action of DEET though should be revised. "There may also be additional mechanisms for the action of DEET...", should read "There are, however, additional mechanisms through which DEET function in *Aedes*", or something along those lines.

We thank the reviewer for checking our revised manuscript and providing a positive appraisal of the work. We have changed the language in the sentence noted by the reviewer. We now write, "*Previous studies have also reported additional mechanisms through which DEET functions in A. aegypti...*"

Reviewer #2 (Remarks to the Author):

Singh et al "Combinatorial encoding of odors in the mosquito antennal lobe"
NComm Review of Revised Manuscript

As was the case for the initial submission this report is an interesting and generally well-presented neuronal examination of the antennal lobe (AL) of *Aedes aegypti* that has now been significantly improved upon revision.

As before, there is much to like here and the authors have generally (apart from ongoing concerns, detailed below) responded to previous issues. Rather than detail (once again) those positives I shall instead focus on areas that remain significantly concerning.

We thank the reviewer for providing another round of feedback on our manuscript. It is encouraging for our team to hear that the reviewer found our work well-presented and improved upon revision. We have made our best attempt to address the additional concerns raised by the reviewer, as detailed below.

1. As in my initial review, I remain deeply concerned with the near complete use of a single (relatively, high) concentration for odorants used throughout this study. The revision now provides what can only be described as a token gesture of 1 additional concentration for 'some' odorants (3) and 2 additional concentrations for a single odorant (1oct3ol). This concern is exacerbated as it appears those additional odor concentrations were not actually utilized very much at all as the vast majority of PN/LN recordings displayed in Figures 4 and 5, display a grey value which, according to the legend "indicates that the odor was not tested..."

It is not uncommon in the field of insect olfaction to use similar concentrations of odors as used in our study. For instance, Quadry et al, Nature 2013 (Wilson lab) has used 0.1 and 0.01 dilutions; Knaden et al, Cell Reports 2012 (Hansson lab) used many odors at dilutions of 0.1 or 0.001; Ghaninia et al, European Journal of Neuroscience, 2007 (Ignell lab) used 0.1

concentration of propionic acid and 4-methylcyclohexanol; Ai et al, Nature 2010 (Suh lab) used 0.1 dilution of various acidic odors.

We are saddened by the phrase “token gesture” used by the reviewer. Even at the lower concentrations of the odorants, we have provided data for a considerable number of cells: Cyclopentanone .01 in 53 PNs and 11 LNs, Nonanal .01 in 42 PNs, Geranyl Acetate .01 in 18 PNs and 1-octen-3-ol .001 in 15 PNs. In absolute terms, these numbers are not small at all (many papers in the field based on intracellular recordings report fewer cells). All these data are included in Figures 4b and 5b (non-grey values). They do appear sparse (mostly grey rows), as the reviewer has noted for Figure 4b and 5b, only because we have much larger numbers of cells for the higher concentrations.

The existing Figure 4b included data from 144 uniglomerular PNs whose glomerulus was identified from the morphological figures and were tested with at least 5 odors. Now, we have also included a new Supplementary Figure 3a (reproduced below in low resolution), which shows data from all PNs, including PNs belonging to unknown glomeruli or multiple glomeruli, and with no restriction on the number of odors tested (only 7 out of 208 PNs that did not have spiking responses for any odor were excluded). In this expanded dataset, Cyclopentanone .01 is tested in 60 PNs, Nonanal .01 in 49 PNs, and Geranyl Acetate .01 in 21 PNs.

New Supplementary Figure 3a, including data from all PNs

The dataset presented in this study has been generated by 6-year efforts of two PhD students (one has already completed the PhD and moved on to a post-doc position, and the other one has also submitted the PhD thesis for review). New experiments to collect data for more odorant concentrations will require another multi-year PhD-level effort and are neither feasible nor necessary for the claims that we are making in the manuscript. We have only drawn conclusions that are supported by the data at hand. We have now added a qualifying remark in the first paragraph of the Discussion section:

“However, one limitation of in-vivo intracellular electrophysiology for small neurons is that it requires a very delicate preparation (including removal of a part of the cuticle and peri-neural sheath above the brain), and the slightest vibration can derail the experiment; to record

about 250 cells, we had to go through nearly 1250 mosquito preparations. The practical challenges also limit the recording time and thus the number of odorants tested per cell. And because we wanted to test each cell with different types of odorants, many of the odorants could be tested at a single concentration only. Thus, for assessing dose-response relationships and the neural representations at a larger range of dilutions, future studies that test multiple concentrations of specific odorants will be required.”

2. I am also concerned that the ePhys studies seem (perhaps I am not reading it correctly) to have not been normalized/corrected for solvent (mineral oil/water) related responses. From lines 701-704, it appears that only background using air from a ‘clean empty’ vial (Line 614) was used for data analyses.

We had normalized each odor response with respect to the background response (i.e., the spontaneous neural activity in the 2-second period before the odor delivery). We also compared the odor responses to the solvent responses manually when interpreting the responses (solvent data are shown in Figures 4b and 5b), but we did not want to do a double normalization by automatically subtracting the solvent (mineral oil) responses, for two reasons:

1. We did not observe statistically reliable responses to the solvent in most cases we examined. We had 53 PNs, belonging to 23 glomeruli, in which the mineral oil responses were tested. According to the statistical criterion that we have used for classifying a cell-odor pair as ‘responding’ and ‘non-responding’, we found that PNs belonging to 20 out of the 23 glomeruli were not responding to mineral oil; in 2 glomeruli, only one of the many PNs belonging to the glomerulus were responding; and only in 1 glomerulus (I-MD3), we saw responses to mineral oil in multiple PNs. Thus, as PNs mostly did not respond to mineral oil but only had noise-level responses, subtracting these responses would only increase noise in the data.

2. Even in rare cases where a cell responds to the solvent and the odorant independently, subtracting the solvent response may give a wrong picture as it has been well-established in the field of olfaction that responses to odor mixtures are not always linearly additive. In fact, I-MD3 PNs that responded to mineral oil also responded to most of the odorants, including those odorants that were not diluted in any solvent (had only clean air dilution). Thus, I-MD3 PNs seem to be independently responding to most odorants and to mineral oil (or trace impurities present in mineral oil) and subtracting the mineral oil response from the odorant response will provide an unfairly reduced response for the diluted odorants.

We have now performed an additional analysis by including the mineral oil responses in the statistical criterion used for determining if a PN-odor pair is ‘responding’ or ‘non-responding’. We have taken the set of PNs in which mineral oil was tested and analyzed their responses for odorants that were diluted in mineral oil. With the new criterion, we call a PN ‘responding’ if the response to the odor (analyzed in two 1-s bins) is different from the response to the mineral oil in corresponding time bins. The graph below shows the fraction of PNs activated by each odorant using this new statistical criterion. These fractions are not very different from the fractions obtained using the original criterion in Figure 7a.

Fractions of PNs activated by different odorants using a statistical criterion comparing the odorant response to the mineral oil response. The number above each bar shows the number of PNs in which both the corresponding odorant and mineral oil were tested. Odorants not diluted in mineral oil are not included.

This new graph is now included as Supplementary Figure 8 in the revised manuscript.

3. Despite the additional information in this revised submission, I remain skeptical about the value of the behavioral setup used here. The airflow (5L/min from each side!) was significantly higher than in ePhys (2L/min). It's strange from the PID pulses provided in supplemental data there doesn't appear to be any mixing—was this tube placed just at the odor delivery point??

To maintain the same concentration of odor in the behavioral and the electrophysiological setups, it was actually necessary to use different air flow rates (to compensate for differences in the physical dimensions of the two setups). With the chosen airflow rates, the concentrations of odor reaching the animal were indeed comparable in the two setups as confirmed by the PID recordings (Supplementary Figure 9b and 9d).

We have now included new schematics of the odor delivery systems used in the electrophysiological and behavioral setups (Supplementary Figure 9a and 9c, also reproduced below).

The PID measurement shown previously was taken close to the end of the arm. We have now included an additional PID measurement obtained by placing the tube slightly away from the center towards the non-odorized side but with tube opening pointing to the odorized side. The new graph (included in Supplementary Figure 9d, as trace B) shows that the PID response intensity becomes very low as one goes slightly away from the center towards the non-odorized side, indicating that the odor flow was being maintained in the behavioral chamber as we had intended with minimal mixing between the two sides.

Revised Supplementary Figure 9

4. Even with those flaws it is hardly surprising to find a correlation between behavioral and ePhys data for PNs, I was more interested to see if there was a correlation (or not) to LN activity which was neither presented nor discussed??

We had focused on PNs, as PNs are the only output neurons of the antennal lobe that convey odor information to the higher brain areas. We did not expect the LN population activity to be strongly correlated with behavior, as LNs are expected to play roles in normalizing and decorrelating the activity within the antennal lobe and not convey odor information to any other area.

Based on the reviewer's suggestion, we have now performed this analysis for LNs by comparing the behavioural similarity and LN response similarity for odor pairs. We included all possible pairs of odors with at least 10 LNs in common. For each pair of odors, we calculated the difference in their PIs and the difference in their LN responses (calculated as 1 minus the correlation between the vectors of odor-evoked responses in LNs that were tested with both odors). Over all pairs of odors, we did not find any correlation between difference in PI and difference in LN responses.

We have now included this plot as Supplementary figure 10b and reported it in the Results section: “*This correlation was specific to PNs, as a similar analysis performed with LNs revealed no such correlation ($R = 0.013$, $P = 0.9$, $n = 90$ odor pairs from 14 odors; Supplementary Figure 10b).*”

Overall, I remain underwhelmed by these concerns regarding the experimental rigor of this otherwise intriguing report.

We again thank the reviewer for providing additional feedback and hope that the reviewer will appreciate our sincere effort in addressing the concerns, as discussed above for each point.

Reviewer #3 (Remarks to the Author):

I have even more enthusiasm for the manuscript after the the authors' revision. I appreciate that the concerns I had about some aspects of the manuscript have been appropriately addressed. The manuscript is now ready for publication and will make a significant contribution to our understanding of mosquito olfaction.

We thank the reviewer for checking the revised manuscript and recommending the publication of the manuscript.

REVIEWER COMMENTS

Reviewer #4 (Remarks to the Author):

This interesting manuscript entitled “Combinatorial encoding of odors in the mosquito antennal lobe” describes an electrophysiological survey of odor responses in PNs and LNs in the *Aedes aegypti* antennal lobes. This is an important topic and the data provide a foundation for mosquito olfaction research. I am certainly in favor of this work being published in Nature communication. However, although I am aware that this is a revised manuscript, this is the first time I have reviewed the work. I therefore have a number of suggestions that I think the author’s should attend to and that I hope might clarify some of the manuscript:

I will raise the concerns line by line as they appear in the manuscript.

1. Line 112. The authors discuss their preparation for recordings but do not state whether they are recording from female or male *Aedes*. It’s never explicitly stated in the main text what sex is used and I only assumed it was females because these are the blood feeders and on line 146 they eventually inform the reader that the filled neuron traces were registered to a female *Aedes* brain template. This needs to be clarified as early as possible.

2. Lines 67-101, the authors start two long paragraphs that appear to argue against labeled line coding in mosquito olfaction. This is interesting and important for the context of the work but it is surely possible that both combinatorial and labeled line coding is employed. For example, the authors discuss labelled lines for pheromones in other insects so what about mosquito pheromones? I understand that they are stressing that the host odor is complex and so host detection is likely to involve multiple channels, but that doesn’t negate a few of them being ‘labeled’. I recognize that this point is a motivating factor in their study because I think they conclude that host odors are not uniglomerular in their activation. For this reason, a little adjustment of these paragraphs would better set up their argument/motivate the work that follows.

3. Line-151, the Authors mention that PN innervation at the LH was more extensive than in the calyx. What do they mean by that? and why is that important? Is this specific for *Aedes*? Or the same as *Drosophila*? Are they suggesting that *Aedes* might be impaired in learning about the odors represented by these PNs?

4. Similarly on line 158 the authors state that PNs innervating dorsomedial glomeruli with AL mostly project to dorsal LH and PNs from other glomeruli go to ventral-anterior LH. What is the point being made here by this anatomical separation? Are they suggesting something akin to the fruit/pheromone distinction for example drawn in *Drosophila* by the Jefferis group?

5. Line 176-line 179. The authors talk about relative LN innervation of various glomeruli but they don’t mention the potential weakness of the data (they have only sampled some of the PNs and LNs and so

others from different developmental cluster/clonal units might innervate these glomeruli) until the discussion. I think all of these caveats/points that relate to the data collected need to be brought forward from the discussion into the relevant results sections so that readers are aware that the authors know this, while they are reading each section of the paper. Otherwise, readers are likely to unfairly discredit the authors as they read the script. Imagine the reader loses focus and never made it to reading the discussion! I'd imagine the authors would like to avoid such a scenario.

6. Line-186. Based on co-filling of multiple neurons after patch-recording from one, the authors speculate about the presence of gap junctions in the AL. This is interesting and it would help if the authors were more clear about which neurons they identified filling between, PN-PN LN-LN, and/or PN-LN?

7. Line-194. The authors mention spikes and bursts (illustrated nicely in Figure 3a) but seem to only consider isolated spikes for further study. Is there a specific reason for this? If so, it would be worth mentioning in the text. Also, I couldn't see any reference to how 'isolated spikes' were selected beyond them being single spikes. Although it is not clear, I think the trace in 3a is odor-evoked but the trace and legend do not contain information about when the odor is presented and what the odor identity is. I also cannot see mention at this stage whether the authors observed spontaneous spiking (ie. before odor presented, I assume so especially given the traces in Figure 4a where some neurons can even be seen to burst before odor is presented) in these neurons and how they would differentiate between spontaneous isolated spikes and odor-evoked events. Also, it seems critical that they also define how they determine/classify that something is odor-evoked. It seems counter intuitive to not analyze odor-evoked bursting. Again, the authors should explain in the text why they have only analyzed the isolated spikes in their data.

8. In line 191- 215. Electrophysiological classification. Although the authors made great effort to describe how their data look, they do not explain the meaning of each classification, for example "X" number of PN shows the properties of large spike half-widths. They should provide an interpretation of what does that mean. What do these features suggest about those classes of neurons? What is the functional importance of these distinctions? Perhaps the authors can cite some old literature and explain to the reader what these data mean and how they might set the bar higher for the mosquito olfaction field, and perhaps also the invertebrate-electrophysiology field.

9. Line 231. The authors mention taking 14 odors and characterizing them into human/plant but it's perhaps also important to characterize them on the basis of functional groups of chemicals such as phenol/ethers/esters? If not, the authors should say why it's more important to define them as human/plant. Perhaps a table listing chemical properties would be of additional help to potential readers.

10. Line 241. The authors mention that PNs exhibit bouts of excitation and inhibition or mixed events but again details appear to be missing. Are these responses specific to any particular chemical compound type? Or specific to morphological PN type? And what does such a result mean?

11. Line 247. The authors mention delayed responses to some odors but they do not explicitly state how they define a delayed response as being stimulus evoked, provide any logic why some neurons might show a delayed response, and what the functional significance of this might mean. Again, have the authors considered whether a particular morphological PN type more often shows delayed responses? Or neurons showing the same electrophysiological properties show delayed responses? Or is this more general? This type of information should be provided. In the same context, the authors mention in supplementary figure 3 (b) that these delays depend on specific PN-odor combinations, but in this sup fig 3b, the authors used two odors, one a ketone and one an alcohol, do they have other examples for they might add in the supplement 3(b) to make this statement a bit more convincing?

12. Line 267. The authors discuss that their recordings indicate that PNs innervating the same glomerulus sometimes appear to be differentially tuned. However, they note that this could be due to the fact that they usually record from multiple Aedes to draw this conclusion. They note that this could be a confound of between animal recordings but not what differences between animals might account for this. Do they mean that the ORN-glomerulus connectivity might vary between animals? I think it is equally possible that the authors misattribute the PNs to a particular glomerulus at times. If this is not likely the authors should say why.

13. Line 273. The authors mention that PNs from dorsomedial glomeruli show a bias towards sending axon projections to dorsal region within LH and that they respond to odor similarly. Again, they describe their findings very generally but do not address what it means.

Few minor comments:

1. What is the estimated number of PNs and LNs in each AL? And are the numbers likely to vary between animals?
2. Do the authors maintain the same temperature during the e-phys experiments and behavior experiments?
3. Why do the authors only use mated females for the experiments? Is the behavior also only from females or mixed populations? (I understand why it would only be females but it's not clearly stated. It would perhaps be worth a few lines in the discussion to mention whether the authors would expect different results if they had used unmated females, non-blood-fed-mated females, or even males.

Reviewer #4 (Remarks to the Author):

This interesting manuscript entitled “Combinatorial encoding of odors in the mosquito antennal lobe” describes an electrophysiological survey of odor responses in PNs and LNs in the *Aedes aegypti* antennal lobes. This is an important topic and the data provide a foundation for mosquito olfaction research. I am certainly in favor of this work being published in Nature communication. However, although I am aware that this is a revised manuscript, this is the first time I have reviewed the work. I therefore have a number of suggestions that I think the author’s should attend to and that I hope might clarify some of the manuscript:

We thank the reviewer for highlighting the importance of our work and recommending its publication in Nature Communications. We are grateful to the reviewer for various suggestions for improving the clarity, which we have incorporated as described below.

I will raise the concerns line by line as they appear in the manuscript.

1. Line 112. The authors discuss their preparation for recordings but do not state whether they are recording from female or male *Aedes*. It’s never explicitly stated in the main text what sex is used and I only assumed it was females because these are the blood feeders and on line 146 they eventually inform the reader that the filled neuron traces were registered to a female *Aedes* brain template. This needs to be clarified as early as possible.

Yes, only females were used. We have now mentioned “female *Aedes aegypti*” in the first line of the Results.

2. Lines 67-101, the authors start two long paragraphs that appear to argue against labeled line coding in mosquito olfaction. This is interesting and important for the context of the work but it is surely possible that both combinatorial and labeled line coding is employed. For example, the authors discuss labelled lines for pheromones in other insects so what about mosquito pheromones? I understand that they are stressing that the host odor is complex and so host detection is likely to involve multiple channels, but that doesn’t negate a few of them being ‘labeled’. I recognize that this point is a motivating factor in their study because I think they conclude that host odors are not uniglomerular in their activation. For this reason, a little adjustment of these paragraphs would better set up their argument/motivate the work that follows.

We did not comment on the detection of pheromones in the mosquito brain as no prior data is available on their detection in the antennal lobe. We have now added the possibility of a few labeled lines in addition to a combinatorial code. We now write, “*While it is possible that a select few of these components or other pheromones in mosquitoes are detected by labeled lines, it seems unlikely that each of these components of the human odor has a dedicated neural pathway for itself in the mosquito brain*”.

3. Line-151, the Authors mention that PN innervation at the LH was more extensive than in the calyx. What do they mean by that? and why is that important? Is this specific for *Aedes*? Or the same as *Drosophila*? Are they suggesting that *Aedes* might be impaired in learning about the odors represented by these PNs?

We have clarified that there was more branching in the LH compared to MB, and that this is also the case in *Drosophila*. So, no functional difference from *Drosophila* is being implied.

We now write, "*PN innervation at the lateral horn was in general more extensive (more branches) than in the calyx, as previously reported in Drosophila (Stocker et al., 1990).*"

4. Similarly on line 158 the authors state that PNs innervating dorsomedial glomeruli with AL mostly project to dorsal LH and PNs from other glomeruli go to ventral-anterior LH. What is the point being made here by this anatomical separation? Are they suggesting something akin to the fruit/pheromone distinction for example drawn in *Drosophila* by the Jefferis group?

We have looked at the functional implication of this anatomical separation later in the "Odor responses of projection neurons" section of the Results and Supplementary Figure 4d. In mosquitoes, the distinction is not very clear like the fruit/pheromone distinction in *Drosophila*. We only conclude that the "*PNs innervating the dorsomedial glomeruli appear to be slightly more similar to each other morphologically and functionally than other PNs.*"

5. Line 176-line 179. The authors talk about relative LN innervation of various glomeruli but they don't mention the potential weakness of the data (they have only sampled some of the PNs and LNs and so others from different developmental cluster/clonal units might innervate these glomeruli) until the discussion. I think all of these caveats/points that relate to the data collected need to be brought forward from the discussion into the relevant results sections so that readers are aware that the authors know this, while they are reading each section of the paper. Otherwise, readers are likely to unfairly discredit the authors as they read the script. Imagine the reader loses focus and never made it to reading the discussion! I'd imagine the authors would like to avoid such a scenario.

We have now added the following caveat in the section pointed by the reviewer, "*We note that this result is based on the limited set of 41 LNs, and it is possible that others LNs that were not sampled in our experiments may have stronger innervations in these glomeruli.*"

Additionally, we have now added the following limitation in the first paragraph of the results, "*One limitation of our preparation was that we could not target cell bodies located in the ventral part of the AL.*"

6. Line-186. Based on co-filling of multiple neurons after patch-recording from one, the authors speculate about the presence of gap junctions in the AL. This is interesting and it would help if the authors were more clear about which neurons they identified filling between, PN-PN LN-LN, and/or PN-LN?

We have added the following details in this part: "*In most of these cases, the patched neuron was a PN while the additional cell bodies that got filled included PNs or LNs, suggesting the presence of gap junctions between PN-PN or PN-LN pairs. We do not have sufficient data to comment on LN-LN gap junctions.*"

7. Line-194. The authors mention spikes and bursts (illustrated nicely in Figure 3a) but seem to only consider isolated spikes for further study. Is there a specific reason for this? If so, it would be worth mentioning in the text. Also, I couldn't see any reference to how 'isolated

spikes' were selected beyond them being single spikes. Although it is not clear, I think the trace in 3a is odor-evoked but the trace and legend do not contain information about when the odor is presented and what the odor identity is. I also cannot see mention at this stage whether the authors observed spontaneous spiking (ie. before odor presented, I assume so especially given the traces in Figure 4a where some neurons can even be seen to burst before odor is presented) in these neurons and how they would differentiate between spontaneous isolated spikes and odor-evoked events. Also, it seems critical that they also define how they determine/classify that something is odor-evoked. It seems counter intuitive to not analyze odor-evoked bursting. Again, the authors should explain in the text why they have only analyzed the isolated spikes in their data.

The exact criterion for the selection of isolated spikes is provided in the Methods. Based on the reviewer's suggestion, we have now added a reference to Methods just after the first mention of isolated spikes.

The number of burst spikes was indirectly used in our analysis, but other features of burst spikes were not used. We have now added the rationale in the Results section: "*We did not use amplitudes, widths, and after-hyperpolarization amplitudes of burst spikes in this analysis as they were quite variable within a cell; the fraction of burst spikes was implicitly accounted in the fraction of isolated spikes (the two fractions add up to 1)*".

The spikes observed in Fig 3a are not odor-evoked spikes but are spontaneous spikes. Indeed, as mentioned later in the section on "Odor Responses of projection neurons", many PNs showed spontaneous spikes (including bursts). Determining odor-evoked spikes was easy, as the odor was delivered at a specific time (2-3 s) during the 10 s trial. We have provided additional details of how odor-evoked spikes were analyzed in the "Analysis of odor responses" section of Methods. For the purpose of this classification analysis based on electrophysiology features, we did not differentiate between spontaneous activity and odor-evoked events. We wanted to use those electrophysiological features that are properties of the recorded neuron and not of the stimulus. These features would allow one to identify neurons as LNs or PNs independently of the stimulus, and then one can later compare the odor responses of these two types of neurons. However, if one already uses the information about odor responses in differentiating LNs and PNs, then one cannot use the same neurons to compare the odor responses of the two classes of neurons (as it would become a circular analysis). Keeping this in mind, we used only those electrophysiological features that depend on the fundamental membrane properties of a neuron and should be independent of the stimulus identity or concentration (e.g., the spike amplitude). In our analysis presented in the manuscript, we have extracted these basic electrophysiological features for each neuron using all the recordings available for that neuron. To confirm that the extracted features and the classification are not dependent on odor responses, we performed this analysis again by excluding spikes in the odor response duration from each trial. The plots below show the results for this revised analysis: the values of ephys features in figures **a-d** below are very similar to **Figures 3c-f** in the manuscript. Thus, we are confident that our classification based on ephys features is independent of the stimulus, as we intended it to be. This analysis has now been included as Supplementary Ffigure 2 a-d, and explained in the Results.

8. In line 191- 215. Electrophysiological classification. Although the authors made great effort to describe how their data look, they do not explain the meaning of each classification, for example “X” number of PN shows the properties of large spike half-widths. They should provide an interpretation of what does that mean. What do these features suggest about those classes of neurons? What is the functional importance of these distinctions? Perhaps the authors can cite some old literature and explain to the reader what these data mean and how they might set the bar higher for the mosquito olfaction field, and perhaps also the invertebrate-electrophysiology field.

The rationale for doing this analysis is: (1) it confirms that PNs and LNs have systematic differences in the membrane properties; and (2) it provides a simple way for future studies to identify whether a recorded neuron is a PN or an LN, even if the morphological information is not available for that neuron (in cases where dye-fills are not available or successful, or in cases where multiple neurons are recorded in the same antennal lobe and it is not possible to identify them separately using just one or two fluorescent dyes that may be available for use).

Based on this analysis, we have shown the differences in the properties of PNs and LNs in Figure 3c-f. These differences likely arise from different shapes and channels in these types of neurons, but in the absence of further experimentation, it is not possible for us to comment on what channels may be involved in generating these differences. At present, we have focused on the practical utility of being able to assign identity to a neuron without dye-filling, with 95% accuracy. We had reported the properties of the remaining 5% neurons for understanding why they were misclassified.

We now write, “*The high classification accuracy confirms that PNs and LNs have systematic differences in their electrophysiological properties, and that it is possible to use these differences to identify a recorded AL neuron as an LN or a PN even without knowing the morphology of the neuron (e.g., in the absence of dye-fills). We observed that out of 212 neurons analyzed, only 2 PNs and 6 LNs were placed in the wrong clusters. We looked further into these specific neurons to see why they were misclassified and found that the 2 misclassified PNs had unusually large fractions of isolated spikes (0.80 and 0.56 respectively, compared to the group mean of 0.04), while the misclassified LNs had unusually large spike half-widths (2 cases), small after-hyperpolarization (1 case), or small spike amplitudes (3 cases). Further, we checked if the electrophysiological properties depend on the morphological sub-classes within PNs and LNs (physiological sub-classes of local neurons have been previously observed in cockroaches (Husch et al, Journal of Neurophysiology, 2009))...*”

9. Line 231. The authors mention taking 14 odors and characterizing them into human/plant but it's perhaps also important to characterize them on the basis of functional groups of chemicals such as phenol/ethers/esters? If not, the authors should say why it's more important to define them as human/plant. Perhaps a table listing chemical properties would be of additional help to potential readers.

We thank the reviewer for this suggestion. We have now added a new Supplementary Table 1, showing the functional groups of 14 odorants used in our study.

10. Line 241. The authors mention that PNs exhibit bouts of excitation and inhibition or mixed events but again details appear to be missing. Are these responses specific to any particular chemical compound type? Or specific to morphological PN type? And what does such a result mean?

We have clarified this in the Results. We now write, "*These temporal patterns were not specific to any odor or any PN but depended on the PN-odor combination.*"

At the end of the paragraph, we have added a new sentence to indicate the significance of these observations: "*These observations of PNs responding to multiple odors with different temporal patterns, including odor-specific onset delays and durations, point to a rich spatiotemporal code for odors.*"

These results set the stage for the analysis of the spatiotemporal code (Figure 7).

11. Line 247. The authors mention delayed responses to some odors but they do not explicitly state how they define a delayed response as being stimulus evoked, provide any logic why some neurons might show a delayed response, and what the functional significance of this might mean. Again, have the authors considered whether a particular morphological PN type more often shows delayed responses? Or neurons showing the same electrophysiological properties show delayed responses? Or is this more general? This type of information should be provided. In the same context, the authors mention in supplementary figure 3 (b) that these delays depend on specific PN-odor combinations, but in this sup fig 3b, the authors used two odors, one a ketone and one an alcohol, do they have other examples for they might add in the supplement 3(b) to make this statement a bit more convincing?

We have expanded the description of these results to address the questions raised by the reviewer. In particular, we have emphasized that the onset delays are not specific to PN types, but vary with the odor for given a PN. We have also included a new figure panel (Supplementary Figure 3c) to show more examples of the onset delays for different PN-odor combinations.

In the Results, we now write, "*Usually, the response started within 150 ms of the triggering of the odor delivery (this delay includes the time it took for the odor vapors to travel to the animal). However, some odor responses in PNs had longer onset delays of >500 ms, which were consistent across different trials. These delays in the onset of responses cannot be attributed to delays in the delivery of specific odors as we observed that the same odor that generated a delayed response in one PN could generate a fast response in another PN (Supplementary Figure 3b). A given PN could respond with a small delay for one odor and a large delay for another odor; thus, the onset delays depended on specific PN-odor combinations (Supplementary Figure 3b and 3c). These variable delays in the onset of spiking in a PN likely reflect odor-specific inhibition received by the PNs through lateral*

inputs. These observations of PNs responding to multiple odors with different temporal patterns, including odor-specific onset delays and durations, point to a rich spatiotemporal code for odors.”

12. Line 267. The authors discuss that their recordings indicate that PNs innervating the same glomerulus sometimes appear to be differentially tuned. However, they note that this could be due to the fact that they usually record from multiple *Aedes* to draw this conclusion. They note that this could be a confound of between animal recordings but not what differences between animals might account for this. Do they mean that the ORN-glomerulus connectivity might vary between animals? I think it is equally possible that the authors misattribute the PNs to a particular glomerulus at times. If this is not likely the authors should say why.

Some homotypic PN groups (such as I-AL3, I-AM1, I-AM4, I-AC2) show a high correlation in their odor responses while others do not (**Supplementary Figure 4a**). We have mentioned two possible reasons for this spread in the text: differences between differential inputs received by different PNs innervating the same glomerulus (indicating functional diversity among homotypic PNs in mosquitoes) or the differences between the individuals from which the PN recordings were obtained. There are many possible sources of variation across individuals: ORN numbers, OR expression levels, ORN-PN connectivity, PN numbers, etc.

As suggested by the reviewer, another possible reason could have been errors in the identification of glomeruli, but we can rule out this as an explanation of the observed variability in odor responses to a large extent. Two members of our team had independently labeled each histological image. They independently assigned a confidence score (0-5) to each PN based on comparing the location, the shape, the size, and the neighborhood of the labeled glomerulus with the atlas (Ignell et al. 2005). Supplementary Figure 4b includes data from 64 uniglomerular PNs (including 74 homotypic PN pairs) that were morphologically identified. To test if incorrect glomeruli identification is the reason for the observed variability among homotypic PNs, we repeated the analysis by requiring at least 3.5 average confidence score from the two experimenters: this resulted in a smaller but more reliable set with 51 PNs and 39 homotypic pairs (about half of the original set of homotypic pairs). The plot below (now included in the paper as Supplementary figure 4c) shows the correlations between odor responses among these homotypic PN pairs (mean \pm SD: 0.19 ± 0.45). We still see the same level of variability as in the original figure. Thus, removing half of the data with relatively less confident glomerular identifications did not reduce the variability among homotypic PNs. We have added this in the Results.

13. Line 273. The authors mention that PNs from dorsomedial glomeruli show a bias towards sending axon projections to dorsal region within LH and that they respond to odor similarly. Again, they describe their findings very generally but do not address what it means.

This is related to point #4 above. Although we do see that dorsomedial PNs are similar in their odor responses to each other than other PNs, no clear segregation was seen in terms of odors detected by the two groups (atleast among the few odors tested in this study). We do not want to overstate this observation: it is possible that future studies with a larger panel of odors may be able to find clear segregation for some odors.

Few minor comments:

1. What is the estimated number of PNs and LNs in each AL? And are the numbers likely to vary between animals?

In the absence of genetic lines labeling PNs and LNs, or electron microscopy imaging, it is not possible currently to comment on the number of PNs and LNs in mosquitoes (neither from our work nor from any previous study).

2. Do the authors maintain the same temperature during the e-phys experiments and behavior experiments?

Yes, both types of experiments were performed at room temperature. This information is now added in the Methods section.

3. Why do the authors only use mated females for the experiments? Is the behavior also only from females or mixed populations? (I understand why it would only be females but it's not clearly stated. It would perhaps be worth a few lines in the discussion to mention whether the authors would expect different results if they had used unmated females, non-blood-fed-mated females, or even males.

Yes, the behavioral experiments were also performed on female mosquitoes. We have now clarified this in the Results.

It is known from previous studies that mosquito behavior (and therefore neural responses) is dependent on the internal state. We chose mated, non-blood-fed females as these most likely to bite humans. As suggested by the reviewer, we have added the following in the Discussion, *“In the current study, we performed experiments on mated but non-blood-fed female mosquitoes, which need blood from hosts. As the olfactory responses depend on the internal state of the animal (Takken et al., 2001; Davis, 1984), we expect that odor responses of PNs and LNs might change for unmated or blood-fed females and for males.”*

REVIEWERS' COMMENTS

Reviewer #4 (Remarks to the Author):

I apologise to the authors for the delayed response in getting to their revision. I am very happy with the discussion provided and the changes made to their manuscript. I believe these changes have improved the script and I congratulate them on this nice piece of work.